# Revealing dominant patterns of aerosols regimes in the lower troposphere and their evolution from preindustrial times to the future in global climate model simulations

Jingmin Li[1], Mattia Righi[1], Johannes Hendricks[1], Christof G. Beer[1], Ulrike Burkhardt[1], Anja Schmidt[1,2,3]

[1]Deutsches Zentrum für Luft- und Raumfahrt (DLR), Institut für Physik der Atmosphäre, Oberpfaffenhofen, Germany
[2]Meteorological Institute, Ludwig Maximilian University of Munich, Munich, Germany
[3]Yusuf Hamied Department of Chemistry, University of Cambridge, Cambridge, United Kingdom

*Correspondence to*: Jingmin Li (Jingmin.Li@dlr.de)

**Abstract.** Aerosols play an important role in the Earth system, but their impact on cloud properties and the resulting radiative forcing of climate remains highly uncertain. The large temporal and spatial variability of a number of aerosols properties and the choice of different 'pre-industrial' reference years prevent a concise understanding of their impacts on clouds and radiation. In this study, we characterize the spatial patterns and long-term evolution of lower tropospheric aerosols (in terms of regimes) by clustering multiple instead of single aerosol properties from preindustrial times to the year 2050 under three different Shared Socioeconomic Pathway (SSP) scenarios. The clustering is based on a combination of statistic-based machine learning algorithms and output from emissions-driven global aerosol model simulations, which do not consider the effects of climate change. Our analysis suggests that in comparison with the present-day case, lower tropospheric aerosol regimes during preindustrial times are mostly represented by regimes of comparatively clean conditions whereby marked differences between the years 1750 and 1850 emerge due to the growing influence of agriculture and other anthropogenic activities in 1850. Key aspects of the spatial distribution and extent of the aerosol regimes identified in year 2050 differ compared to pre-industrial and present-day, with significant variations resulting from the emission scenario investigated. In 2050, the low emission SSP1-1.9 scenario is the only scenario where the spatial distribution and extent of the aerosol regimes very closely resembles preindustrial conditions whereby the similarity is greater compared to 1850 than 1750. The aerosol regimes for 2050 under SSP3-7.0 closely resemble present-day conditions, but there are some notable regional differences: developed countries tend to shift towards cleaner conditions in future, while the opposite is the case for developing countries. The aerosol regimes for 2050 under SSP2-4.5 represent an intermediate stage between preindustrial times and present-day. Further analysis indicates a north/south difference in the clean background regime during preindustrial times, and close resemblance of pre-industrial aerosol conditions in the marine regime to present-day conditions in the Southern Hemispheric ocean. Not considering the effects of climate change is expected to cause uncertainties in the size and extent of the identified aerosol regimes but not the general regime patterns, due to a dominating influence of emissions rather than climate change in most cases. The approach

and findings of this study can be used for designing targeted measurements of different preindustrial-like conditions, and for tailored air pollution mitigation measures in specific regions.

## 1 Introduction

Aerosols play an important role in the Earth's climate, by direct scattering and absorption of solar and terrestrial radiation
(termed aerosol-radiation interactions), and indirect effects due to modifications of cloud microphysical and radiative properties (termed aerosol-cloud interactions) (Boucher et al., 2013). Aerosols influence the climate system in different ways and show a large spatial and temporal variability due to the large variety of aerosol species, mixing states, residence times and size ranges in the atmosphere (e.g. Lauer and Hendricks 2006; Mann et al., 2010; Pringle et al., 2010; Sessions et al., 2015). Aerosols also affect air quality, human health, and visibility as well as regional and global temperature and precipitation
patterns, whereby many of these interactions between aerosols, air quality and climate remain uncertain (Kulmala et al. 2011, Myhre et al 2017, Bellouin et al 2020).

In order to reduce uncertainties on aerosol-climate interactions, it is vital to gain better knowledge of the temporal and spatial evolution of aerosols over time. The preindustrial period is considered as a baseline for radiative forcing calculations, but
different definitions can be found in the literature, using the year 1750 or 1850. Quantifying preindustrial aerosol conditions is essential for evaluating the magnitude of radiative forcing caused by anthropogenic aerosols (Andreae, 2007). Carslaw et al. (2013) investigated uncertainties in present-day aerosol indirect radiative forcing resulting from aerosol-cloud interactions, and demonstrated that 45% of the variance in the magnitude of the aerosol indirect forcing is explained by natural emissions while only 34% is explained by anthropogenic emissions. Therefore, it is important to obtain precise information about
preindustrial aerosol conditions. Gryspeerdt et al. (2023) demonstrated that the uncertainty in aerosol-cloud radiative forcing is driven by clean aerosol conditions, and suggested constraining aerosol properties in clean conditions as an important goal for future observational studies. Efforts have been made to identify regions on Earth with aerosol properties similar to those during the preindustrial period. Based on modeling studies, such preindustrial aerosol conditions (defined as an overall picture of aerosol properties at that time) can be found mostly over present-day oceanic regions in the Southern Hemisphere (Andreae,
2007; Carslaw et al., 2014; Hamilton et al., 2014; McCoy et al., 2020). Hence, aerosol measurements over the remote Southern oceans can be used to constrain aerosol radiative forcing uncertainties (Schmale et al., 2019; Regayre et al., 2020). On the other hand, aerosols play also an important role in future climate projections (IPCC 2021; Quaas et al. 2022). Andreae et al. (2005) suggest that the strong cooling effect induced by aerosols in the past and present climate could decrease in the future, depending on the temporal and spatial evolution of aerosol emissions. Xu et al. (2018) studied the importance of aerosol
scenarios in predicting future heat extremes and demonstrated that aerosols are more important than greenhouse gases in controlling heat extreme statistics over Northern Hemisphere extra-tropical land areas and the duration of heat extremes in the tropics. A further modelling study by Zhao et al. (2019) suggests that future anthropogenic aerosol changes can significantly

affect future heatwave predictions. Li H. et al. (2022) estimated future PM$_{2.5}$ mass concentrations (aggregated mass of particles less than 2.5 μm in diameter) by applying a Random Forest regression method to global atmospheric chemistry model results and CMIP6 multi-model climate projections. Their study suggests that under low- and medium-emission scenarios (SSP1-2.6 and SSP2-4.5) PM$_{2.5}$ mass concentration decreases by about 40% in East Asia, 20−35% in South Asia, and 15−25% in Europe and North America in 2100 compared to present-day, and that the changes are mainly due to the emission reductions. Only in a high-end radiative forcing scenario (SSP 5-8.5), there is a comparable contribution of changes in climate and emissions to future PM$_{2.5}$ changes over many regions on the Earth (e.g. East Asia, South Asia, Europe and North America). Overall, it is therefore essential to classify aerosol properties and hence characterize aerosol conditions under different future emission scenarios compared with pre-industrial times and the present-day.

Here we use the output of the global aerosol-chemistry-climate model EMAC (ECHAM/MESSy Atmospheric Chemistry) equipped with the aerosol microphysical sub-module MADE3 (the third generation of the Modal Aerosol Dynamics model for Europe adapted for global applications), together with statistic-based clustering algorithms to characterize similarities and differences in aerosol properties from preindustrial times to the present-day and the future (year 2050). In contrast to previous studies that primarily investigated preindustrial aerosol conditions based on a single parameter - cloud condensation nuclei (CCN) - within the context of aerosol-cloud interaction (e.g. Andreae 2007, Hamilton et al. 2014), the method proposed here combines multiple aerosol properties and targets different time periods. The aerosol properties considered in our study are derived from EMAC-MADE3 model-simulated quantities and include mass concentrations of black carbon (BC), particulate organic matter (POM), mineral dust, sea salt, aerosol sulfate, ammonium, nitrate, and particle number concentrations in the Aitken and accumulation modes. Analysing all these variables individually across different time periods would result in numerous different aerosol distribution patterns, which makes it impossible to draw universal conclusions. In this context, clustering of the multivariate aerosol data provides us with an aggregated and condensed picture of aerosol regimes and their evolution. In our previous study (Li et al. 2022; hereafter L22), we successfully defined present-day global aerosol regimes and analysed their internal characteristics regarding the individual aerosol properties (e.g., concentrations of specific aerosol species) for each regime, using the unsupervised machine learning algorithm K-means. The lower tropospheric aerosol regimes, as identified in L22, comprise a background regime (occurring in polar regions), two oceanic regimes, with the northern oceanic regime being more polluted than the southern one, two dust regimes, with one being strongly dust dominated and the other representing a mixture of dust and other pollutants, two biomass burning/biogenic regimes, with one comprising fresh aerosol and another one including more aged particles, and three continental regimes including weakly, moderately, and enhanced polluted conditions. In the present study, we use the approach of L22 for defining global aerosol regimes (hereafter termed primary classification), and further extend the procedure to also investigate finer structures within specific aerosol regimes (hereafter termed secondary classification), and we investigate the temporal evolution of the regimes from the preindustrial period to the year 2050 with further developed analyses procedures. The goal of the primary classification is to characterize the variability range and relative difference of aerosol properties across the globe, while the secondary

classification is used to characterize each specific regime in more detail. The clustering method used in L22, however, was designed for a single time slice and cannot be used for different time periods, as it would lead to incomparable regimes due to the different aerosol conditions in different time periods. More specifically, aerosol conditions during preindustrial times do not agree with the present-day, due to additional contributions from anthropogenic emissions. K-means performs the classification based on individual data variances and using an equal variance criterion for classification. Assuming the same number of regimes (K=9) to be generated for present-day and preindustrial time, the variance of preindustrial regimes and their characteristics would be different from the present-day case due to the different values in aerosol datasets. This would lead to an incomparability of regimes for different time slices. For the present study, we therefore additionally include the supervised Random Forest machine learning algorithm (Ho 1995, Breiman 2001). As a supervised method, the Random Forest algorithm can be trained using data for one specific time slice and applied to all other time slices. In this way, all time slices are analysed consistently and the temporal evolution of the aerosol regimes can be studied. We focus here only on the lower troposphere (from the surface to about 700 hPa), where the aerosol regimes are strongly connected to the emission patterns.

Overall, we aim to address the following questions: (1) How do aerosol regimes develop over time, from pre-industrial times (years 1750 and 1850) to the future (year 2050) under different emission scenarios? (2) Where can aerosol conditions similar to preindustrial times be found in the present-day and future? (3) What is the effect of different emission pathways on the development of the aerosol regimes? Answering these questions can help to target measurements for specific aerosol conditions (e.g. preindustrial-like or highly polluted), to supply information to policy makers (e.g. emission mitigation efforts in specific scenarios and their expected effects), to provide important hints for subsequent studies relying on information about the properties and distribution of atmospheric aerosols (e.g. evaluating environmental impacts of particulate matter).

## 2 Methodology

### 2.1 Global model simulations and data

In this study, we investigate the development of global aerosol regimes based on simulations using a global aerosol-chemistry-climate model for four climatological time slices and three future scenarios. These comprise preindustrial times (1750 and 1850), present day (2015), and future (2050) under three different emission scenarios of the Shared Socioeconomic Pathways (SSPs; O'Neill et al., 2017; Gidden et al., 2017): SSP1-1.9, SSP2-4.5 and SSP3-7.0. SSP1-1.9 is a low emission scenario, projecting close to net-zero carbon dioxide ($CO_2$) emissions by 2050 and a temperature rise of 1.4 K by the end of the century. SSP2-4.5 is a middle-of-the-road scenario, with $CO_2$ emissions staying at current level until 2050 and decreasing afterwards, resulting in a temperature rise of 2.7 by 2100. SSP3-7.0 is a pessimistic scenario, projecting a doubling of $CO_2$ levels and 3.6 K temperature rise by 2100 (IPCC, 2021). We investigate both year 1750 and year 1850 preindustrial conditions. The year 1750 is commonly considered as the preindustrial reference case by several studies (e.g. Boucher et al 2013; Hamilton et al. 2014; Hawkins et al 2017; IPCC 2021), while other studies (e.g., Carslaw et al., 2017) argued that the year 1850 should be

used as a preindustrial reference period when considering aerosol radiative forcing because the year 1850 shows marked difference in terms of aerosol emissions compared to 1750. The IPCC Sixth Assessment Report (AR6) uses 1750 as a preindustrial reference to assess radiative forcing, but uses 1850-1900 for other aspects, e.g. surface temperature change. The IPCC Special Report on Global Warming of 1.5°C (IPCC 2022) uses the period 1850–1900 as preindustrial baseline.

The simulations analysed in this study were performed as part of an assessment of the global impact of the emissions of the transport sector on aerosol and climate (Righi et al., 2023), using the ECHAM/MESSy Atmospheric Chemistry (EMAC) general circulation model (Jöckel et al., 2010, 2016) equipped with the aerosol microphysical sub-module MADE3 (Kaiser et al., 2014, 2019). It simulates nine different aerosol species: BC, POM, ammonium, sulfate, nitrate, the sea salt species sodium and chloride, mineral dust and aerosol water. These nine species are distributed into three different mixing states within three size ranges, resulting in a total of nine aerosol modes for each species. The three mixing states include purely soluble particles, particles mainly composed of insoluble material and only very thin soluble coatings (<10% of the total particle mass), and mixed particles consisting of an insoluble core with a soluble coating. The three size modes are Aitken-, accumulation- and coarse mode. MADE3 considers particle microphysical changes due to condensation, coagulation, gas-particle partitioning and new particle formation. MADE3 was shown to be able to properly represent aerosol microphysical processes, with the simulated aerosol properties showing good agreement with observations, as demonstrated in previous studies (Kaiser et al., 2014; Kaiser et al. 2019; Beer et al., 2020; Righi et al., 2020).

All simulations are performed in the T42L41 resolution, corresponding to a grid of 2.8°×2.8° in latitude and longitude and 41 vertical hybrid σ-pressure levels up to 5 hPa. In this study, we only consider changes in the emissions of short-lived species (aerosols and aerosol precursor gases) for the different time slices, while the climate is held constant at present-day conditions. The time slices are simulated for a duration of 10-years and the climatological means of the 10-years simulation are considered for the respective time slices. Due to the complexity and interactions of microphysical processes in the global aerosol climate model, it would be difficult to distinguish the influence of climate change and emission changes if both would be considered at the same time and our major goal is to link the development of global aerosol regimes through time to the emission patterns. Moreover, L22 demonstrated that emissions are the key drivers for global aerosol regimes, especially for the lower troposphere. Additionally, Koffi et al (2010) showed that the effect of emission changes on transport-induced ozone is larger than the effect of climate change, providing a hint that aerosols could be more influenced by emission than climate. For this reason, EMAC is applied in nudged mode, i.e. model dynamics are constrained using ERA-Interim reanalysis data (Dee et al., 2011), more specifically, temperature, wind divergence and vorticity, and the logarithm of the surface pressure for the time period used to constrain the climate (2006-2015). The emission data applied for preindustrial times, present-day conditions, and future scenarios are taken from the CMIP6 inventory (van Marle et al 2017, Hoesly et al 2018, Gidden et al 2019, Feng et al. 2020), including natural and anthropogenic particulate and gas emissions from different sectors such as biomass burning (BB), agricultural waste burning (AWB), land transport, shipping, aviation, and other anthropogenic sources. Since the emissions of

mineral dust and sea salt are wind-driven, they are calculated online for each model time step using the parameterizations by Tegen et al. (2002) and by Guelle et al. (2001), for mineral dust and sea salt, respectively.

The development of aerosol regimes over time is analyzed on a climatological mean basis. Aerosol properties for the machine learning classification tasks are extracted from multi-annual average (10 year) simulation data. As in L22, the clustering algorithm considers seven aerosol properties: aerosol mass concentrations of BC, mineral dust, sea salt, POM, the sum of sulfate, nitrate, and ammonium (SNA), as well as number concentrations of aerosol particles in the Aitken and accumulation

modes. To characterize aerosol regimes in the lower troposphere, the simulated mass and number concentrations are vertically integrated between the terrain-following hybrid sigma pressure levels 41 (corresponding to the surface) to 33 (corresponding to an altitude range from the ground to about 700 hPa).

**2.2 Description of machine learning classification methods in general**

In this study, we apply two statistic-based machine learning algorithms that compare and evaluate the similarity and differences

among different datasets: the unsupervised K-means clustering and the supervised Random Forest classifier.

K-means (MacQueen 1967, Hartigan and Wong 1979) is an unsupervised machine learning clustering algorithm, which has been applied in several recent atmospheric modeling studies (e.g., Borge et al., 2022; Li et al., 2022; Raudesepp and Maljutenko, 2022). Unsupervised methods require no prior classification knowledge, but need a labeling and evaluation of the

results after the classification. K-means can divide an input dataset into a predefined number of clusters using an equal variance criterion. K-means is based on the calculation of Euclidean distance (Spencer, 2013) which computes the distance between two samples in a multi-dimensional space (with the number of dimensions equal to the number of variables). K-means can perform clustering for a predefined number of clusters, but does not provide information about the optimal number of clusters to be used. Many classification evaluation metrics (e.g. Rousseeuw, 1987; Tibshirani et al. 2001) can support the selection of

an optimal number but there is no general solution for all cases. The input dataset should be standardized for K-means classification, to overcome the problem of input variable values spanning different orders of magnitude, which leads to that input variables are not equally weighted (see L22, for details). The choice of an appropriate scaling method depends on the input dataset and on the application (Milligan and Cooper, 1988).

Random Forest is a supervised machine learning method. A detailed description of Random Forest classification can be found in Ho (1995) and Breiman (2001). Briefly, Random Forest requires a preexisting classification of the input data in order to learn decision rules. Random Forest uses an ensemble learning technique that performs the classification by creating an ensemble of decision trees: the so called 'forest'. The majority of votes from all decisions is considered as the final output. Compared to a single decision tree, this type of ensemble classification improves the performance and avoids the tendency of

a single tree to overfit its training dataset (Hastie et al., 2008). Decision trees (Quinlan, 1986; Quinlan, 1987) are the basis of

a Random Forest. Each decision tree is built by using a unique subset of samples randomly collected from the training data. A decision tree simulates intelligent behavior and has the ability to learn. The decision tree is built with a tree structure including root node, decision nodes, and leaf nodes. All nodes are connected with branches. The paths from the root node to leaf nodes are controlled by decision rules. Decision notes contain decisive questions, yes/no-answers of these depicting questions lead

the decision path to different branches. Depending on how the respective sequences of questions were answered in the decision trees and how the respective branches reach the leaf nodes, a set of class labels is generated. Random Forest is among the best supervised learning algorithms in terms of accuracy and performance (Liaw and Wienner, 2002) and has been widely used in aerosols studies (e.g., Wei et al., 2010; Christopoulos et al., 2018; Choi et al., 2021; Kianian et al., 2021; Lee et al., 2021; Yu et al., 2022).

**2.3 Analysis and applied classification approach**

The analysis and classification approach applied in this study is based on the K-Means algorithm as in L22 and further extended by the Random Forest algorithm. The latter is required due to the equal variance criterion implemented in the K-means algorithm (Hartigan and Wong, 1979), which would divide the datasets based on their individual variances and would therefore lead to the identification of incomparable regimes across the different time periods when performing K-means classification

for each time period independently. Furthermore, applying K-means to a combined dataset of all the different time periods would lead to comparable regimes across all time periods, but the classification results change whenever a new time period or scenario is considered. To overcome these limitations, we developed a two-step approach using a combination of K-means and Random Forest, which is outlined schematically in Fig. 1. First, we choose present-day (REF-2015) as our reference time period and apply the L22 procedures to this reference time. Next, we apply the Random Forest classification rules learned

from REF-2015 and apply these rules to all other time periods in order to generate comparable and consistent aerosol regimes. This, in turn, allows us to identify present-day and future aerosol regimes that have a high probability to feature aerosol conditions that are similar to those during the pre-industrial period.

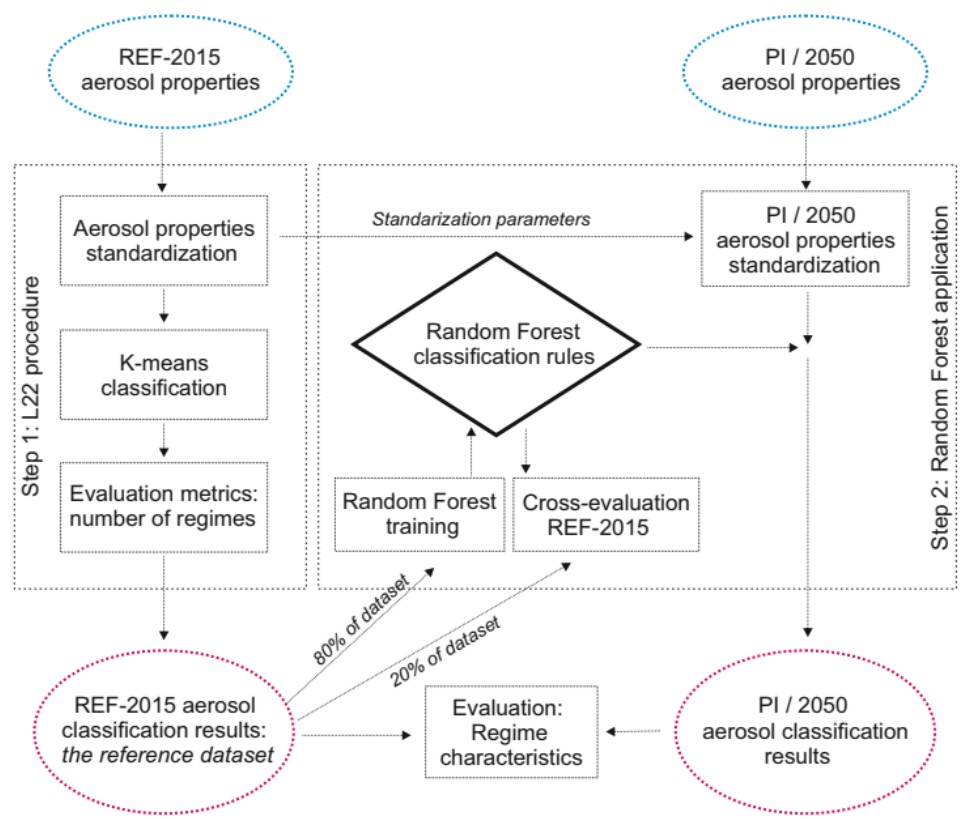

**Figure 1: Workflow of applied machine learning classification procedures. Blue ovals represent the input data. Red ovals show the output data. Rectangles represent procedures to analyze the data. Diamond represents decision rules. Arrows show the sequence of analysis.**

Present-day conditions are selected as our reference for the following reasons. First, our simulations are constrained by present-day climate, so the simulation for the present-day case are likely more accurate than simulations of other time slices. Second, present-day emissions are more reliable than those for past and future conditions. Third, the pre-industrial cases are not fully representative of present-day aerosol conditions, thus using the preindustrial case as a reference, the regime classification for the present-day and 2050 might be incomplete. The global aerosol classification for present-day conditions (REF-2015) follows exactly the same procedures as shown in L22, including aerosol properties standardization, K-means classification, and evaluation for the choice of the final number of regimes. The values of the REF-2015 input aerosol properties are standardized before applying K-means in order to be equally weighted by the algorithm. We use the most suitable standardization method for global aerosol classification, which is based on sample mean and sample standard deviation. More specifically, each aerosol property is standardized by subtracting its sample mean and dividing the difference by its sample standard deviation. K-means classification is conducted for a range of defined numbers of regimes. The choice of the optimal number of aerosol regimes is

based on two evaluation metrics (the sum of squared errors and the silhouette coefficient). The L22 procedure applied to the REF-2015 data generates a sample dataset containing the values of the considered aerosol properties and the assigned regime index for each individual model grid point (a dataset of 8192 points resulting from 128 longitude and 64 latitude grid points). This dataset is considered the 'ground truth' for the following machine learning processes and serves as the training dataset

for the supervised machine learning with the Random Forest classifier in the second step of our procedure.

The Random Forest classifier has the ability to learn how to make decisions for specific aerosol regimes based on the training data generated for REF-2015 using K-means. The input properties for all other time slices are standardized consistently using their corresponding REF-2015 standardization parameters (mean and standard deviation) to create consistent inputs for the

Random Forest algorithm. A cross-validation process is applied to the Random Forest learning by selecting 80% of the REF-2015 data points, which the algorithm uses to learn the aerosol classification decision rules. The evaluation is conducted for the remaining 20% of REF-2015 data points which are excluded from the learning process, by comparing aerosol regime labels generated by the Random Forest decision rules to the original K-means classification. The resulting accuracy of the Random Forest classifier ranges between 94.5% and 98.8%, and is therefore well-suited for the analysis conducted in this study. The

accuracy of the Random Forest algorithm is also evaluated, by comparing the internal properties of the aerosol regimes in REF-2015 generated by K-means and those of the other time slices generated by Random Forest (Sect. 3.1). The above procedures are applied for both the primary and secondary classification for the following reasons: 1) The classification of the full global data set (primary classification) can identify aerosol regimes that, on a global scale, are characterized by marked difference in aerosol properties. In detail, the global data span a wide range of values and aerosol properties, which are

distributed quite heterogeneously over the globe. Due to this large variability in the global dataset, the primary classification identifies regimes with clear difference, e.g. regimes containing high local values for specific aerosol species and regimes where the values of aerosol properties are close to their global minimum (hereafter termed clean aerosol regimes), however it cannot further distinguish differences within the clean regimes. We therefore conduct a secondary classification to investigate fine structures within clean aerosol regimes, using the same procedure as for the primary classification, but applied to a subset

of the data representative of a specific primary aerosol regime. With the secondary classification we zoom into a specific aerosol regime and can identify further detailed differences within it. The secondary classification is of special importance for the analysis of preindustrial regimes or future regimes under low pollution scenarios, such as SSP1-1.9, where clean aerosol regimes dominate.

## 3 Results and discussions

### 3.1 Primary classification of aerosol regimes and their properties

Before discussing the development of primary classification aerosol regimes over time, it is important to distinguish the regime differences caused by numerical artifacts of the algorithm from the real regime changes in the model simulations. First, we

need to evaluate whether the aerosol regimes identified across the other time periods by the Random Forest algorithm agree with the present-day ones as classified by K-means (i.e., the 'ground truth' for Random Forest learning). The comparison of
aerosol properties for the identified primary classification aerosol regimes across time periods (Fig. S1 in the Supplementary Material) shows that similar internal aerosol properties are derived for each identified aerosol regime from primary classification between REF-2015 and the preindustrial and future time periods. This suggests that Random Forest correctly learns the classification criteria and is well suited for this study. One minor discrepancy between the two algorithms is the classification of the dust-dominated regimes: the data distributions of the dust-dominated level 1 regime in the preindustrial
and in SSP1-1.9 is characterized by many outliers compared with the distribution in REF-2015, which should instead be assigned to the dust-dominated level 2 regime. This discrepancy is due to the fact that K-means is based on the Euclidean distance and is therefore more sensitive to the large values of a distribution, while Random Forest is less sensitive to them as it predicts classes in a tree structure. This discrepancy suggests that it would be more appropriate to discuss the two dust-dominated regimes as a single one. Considering also that the dust-dominated level 2 regime is small in terms of sample size,
the differences between these two regimes should not be over-emphasized.

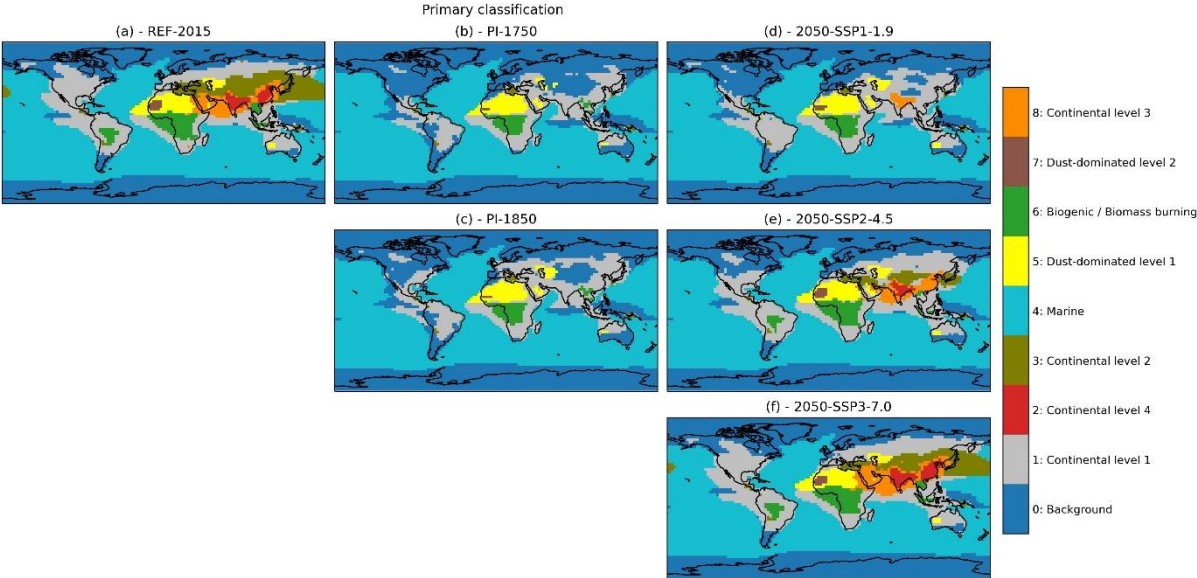

**Figure 2: Present-day (a), preindustrial times (b-c) and future (d-f) aerosol regimes from primary classification. The aerosol properties of each present-day regime are shown in Fig. 3.**

In the following, we first discuss the identified present-day primary classification aerosol regimes (Fig. 2) and their properties (Fig. 3), then discuss these for the preindustrial cases and the future. Fig. 3 shows comprehensive and integrated information on the present-day regime characteristics in terms of individual aerosol properties as classified by means of by K-means. In

our approach, these K-means classification results also serve as a learning criterion for the Random Forest classification for other time periods. The different colors in Fig. 3 represent the different aerosol properties considered in this study. The y-axis shows their standardized values, with a higher (lower) standardized value corresponding to a higher (lower) aerosol mass and number concentration (i.e. the standardization process normalizes values of different aerosol properties to the same order of magnitude, while conserving the underlying distribution of these aerosol properties). The standardized values are also used to define the different pollution levels in the continental and dust-dominated clusters. The Random Forest algorithm learns from value ranges and the relative importance of the considered aerosol properties for each regime (regime characteristic), and then maps the pre-industrial and 2050 aerosol properties to the identified regimes. The same regime identified during pre-industrial times and 2050 represents the same conditions as the present-day regime (evaluated in the Fig. S1). We recall that the simulations analyzed here only consider the impact of changing emissions, while the impact of climate change is neglected. This might affect the size and extent of pre-industrial and future regimes to a certain extent but it should not change the classification substantially, since previous studies suggested a distinctively larger importance of emission changes than climate change for the evolution of the lower tropospheric aerosol (see detailed discussions in Sect. 4). Figures 2a and 3 show that there is a background regime in the polar regions (regime 0), where all aerosol properties show lower values than the other regimes. The marine regime (regime 4) is dominated by an enhancement in sea salt, while all other aerosol properties show only low values. Regimes 5 and 7 are dust-dominated regimes located over the Sahara. Regime 6 can be related to biomass burning activity/biogenic emissions in forest regions and savannas of South America and Africa and the respective downwind areas. In addition, regime 6 is characterized by a clear enhancement in POM, BC and particle number concentration in the accumulation mode, which is typical of this emission source. Four different continental regimes are identified, which are characterized by different levels of aerosol due to anthropogenic influences (Regime 1, 3, 8, and 2, hereafter referred to continental level 1 to level 4 regimes, respectively, in increasing order of anthropogenic influences).

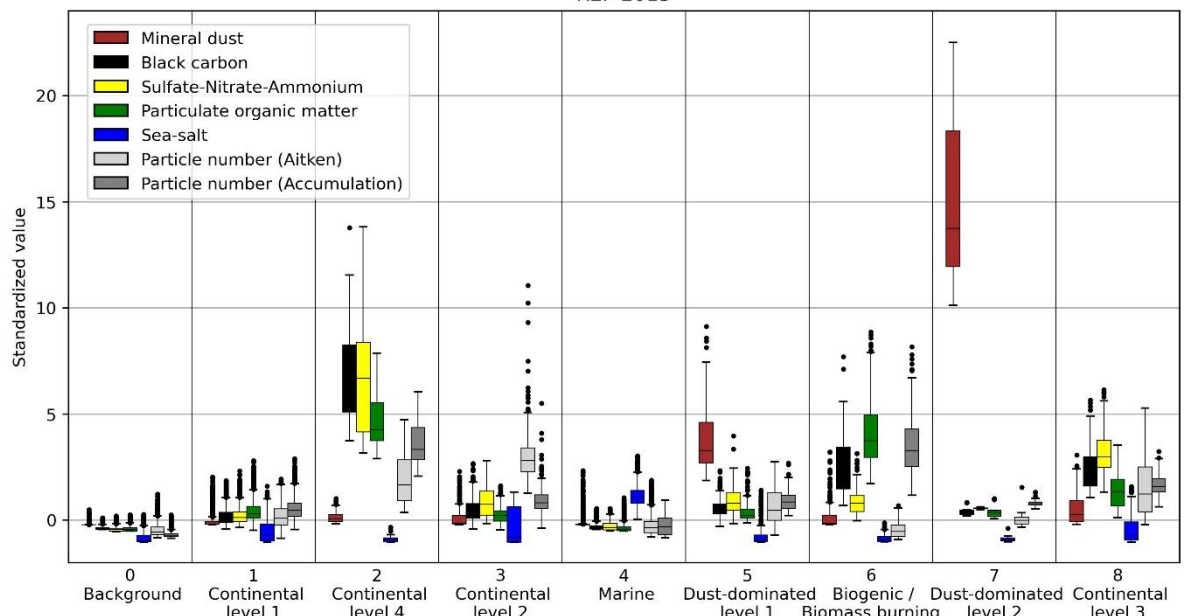

**Figure 3: Internal aerosol properties of the primary classification aerosol regimes shown in Fig. 2. The boxes mark the range of data between the 25% and 75% quantiles, referring to the interquartile range (IQR). The whiskers indicate the minimum and maximum**
**value within the 1.5×IQR distance from the box. Data beyond the whiskers are displayed as dots. The raw input values are standardized by removing the sample mean and dividing by the sample standard deviation, this standardization procedure preserves the data distribution and changes only data values. The standardized value 0 represents the global average of each aerosol property. The negative (positive) standardized values describe data below (above) global average expressed as factors of their global standard deviation.**

The aerosol regimes in preindustrial times (Fig. 2b, c) are mostly characterized by three clean primary classification aerosol regimes (background, marine, and continental level 1) covering the whole globe. The spatial extent of the background regime in preindustrial times is considerably larger than for present-day, covering large parts of the continents in the Northern and Southern Hemisphere. The dust-dominated (regimes 5 and 7) and the biomass burning/biogenic regimes (regime 6) over Africa
exist throughout all time periods, although the spatial extent of the biomass burning/biogenic regime over Africa in preindustrial times and in SSP1-1.9 is smaller than during the present-day. This indicates that the spatial extent of biomass burning / biogenic regime in preindustrial times can be considered as a natural emission baseline, with the anthropogenic contributions at present-day amplifying the biomass burning / biogenic regime to cover a larger area. The biomass burning regime over South America, however, is not present in preindustrial times, suggesting the anthropogenic origin of the biomass
burning activity apart from agriculture in this region. Differences between the primary continental level 1 regimes emerge between the two preindustrial time periods (PI-1750 and PI-1850), due to the growing influence of agriculture and other anthropogenic activities between these two time periods. For example, North America, Europe and northern Asia belong to the background regime (regime 0) in PI-1750, but shift to continental level 1 (regime 1) in PI-1850 with implications for the

choice of a representative year for preindustrial conditions when calculating aerosol radiative forcing (e.g. Carslaw et al. 2017; IPCC 2022).

Key aspects of the spatial distribution and extent of the primary classification aerosol regimes in the future differ compared to pre-industrial and present-day, with significant variations across the three SSP scenarios investigated here. SSP1-1.9 is the scenario under which the primary classification aerosol regimes are most similar to 1750 or 1850. Under the low (SSP1-1.9, Fig. 2d) and moderate (SSP2-4.5, Fig. 2e) emission scenarios, the shift towards preindustrial conditions is evident globally, suggesting a general reduction of aerosol and aerosol precursor emissions in line with the underlying assumptions in these scenarios (van Vuuren et al. 2017; Fricko et al. 2017). The spatial extent of continental level 2 to level 4 regimes (regimes 3, 8, and 2) is reduced under the SSP2-4.5 scenario, and these regimes disappear almost completely under SSP1-1.9. The low emission SSP1-1.9 scenario is the only scenario where the spatial distribution and extent of the primary classification aerosol regimes very closely resembles preindustrial conditions, whereby the agreement is larger for PI-1850 than PI-1750, except for India that still belongs to the continental level 3 regime (regime 8). The scenario with most similarities with the present-day primary classification aerosol regime distribution is SSP3-7.0 (Fig. 2f), however there are some key differences between SSP3-7.0 and REF-2015. Parts of western Europe, for example, shift from the continental level 1 regime (regime 1) to the background regime (regime 0), which can be interpreted as a consequence of emission reduction policies adopted by developed countries in this scenario (Fujimori et al., 2017). Developing countries like China and India, on the contrary, are still characterized by high levels of pollution in 2050, resulting in an increase in the extent of the continental level 3 and level 4 regimes (regime 8 and regime 2, respectively). Our results suggest that SSP2-4.5 and SSP3-7.0 are less probable to feature preindustrial conditions than SSP1-1.9, although preindustrial conditions can be expected in these scenarios for specific regions, like the ones covered by the background regime, marine regime and continental level 1 regime (Fig. 2e, f). It needs to be noted that the impact of climate change is not considered here, the possible influence of climate change effects on the present results is discussed in Sect. 4.

Our results show that close-to-preindustrial conditions exist at present-day and under SSP1-1.9 to a greater spatial extent and preindustrial times show more diverse conditions than previous studies suggested. Possible reasons for these differences are discussed in the following: (1) many previous studies identified preindustrial-like conditions based on CCN concentration. Andreae (2007) suggested that by turning off anthropogenic emissions, the simulated CCN concentration over the continents agree with that over the southern oceans, with the CCN concentration ranging from 50 to 200 cm$^{-3}$. Hamilton et al. (2014) used model simulations to identify present-day atmospheric conditions that resemble PI, by estimating the occurrence of days with similar CCN concentrations between PI-1750 and 2000, suggesting that 90% of unperturbed regions occur in the Southern Hemisphere. In contrast to these studies, we consider an aggregated information from multiple aerosol properties (BC, mineral dust, sea salt, POM, ammonium, sulfate, nitrate, particle number in the Aitken mode, and particle number in the accumulation mode). However, as CCN concentrations are strongly related to some of these aerosol properties, our results can be

qualitatively compared to those from previous CCN-related studies; (2) we are evaluating multi-annual mean aerosol properties, while other studies considered seasonal (Andreae, 2007) and daily variations (Hamilton et al. 2014), which could lead to larger differences between preindustrial and present-day conditions than the differences we find using multi-annual means. Therefore, previous studies might have identified a smaller extent of preindustrial-like conditions than our study because more regions are considered to be different when using daily and seasonal data; (3) our results are based on machine learning algorithms, which allow us to identify and characterize the complex relationships and patterns within the data based on similarities and differences in the global distribution of a multitude of aerosol properties, which is a major methodical difference to the previous studies.

The above results from the primary classification show the ability of the algorithm to extract a clear and condensed picture of the global distribution of aerosol conditions for different time periods and regions. Still, there are features that cannot be captured at the spatial scale of the primary classification. For example, in the L22 study, which used a higher spatial model resolution (i.e. T63, corresponding to 1.9°×1.9° in latitude and longitude) for the input dataset, the K-means algorithm could identify a north-south hemispheric difference for the marine regime, with a higher level of pollution in the Northern Hemisphere, but this difference is not evident in the present study, which uses a lower resolution (T42, 2.8°×2.8°). Such differences, however, would be a key information to constrain aerosol forcing uncertainties (e.g. Carslaw et al. 2014, Hamilton et al. 2014, Regayre et al. 2020). The absence of this north/south contrast suggests that our primary classifications are spatially too coarse and that a finer spatial clustering is required. This is achieved by means of a secondary classification, mostly targeting the regimes with low levels of aerosol identified by the primary classification.

### 3.2 Secondary classification of aerosol regimes and their properties

For the secondary classification shown in Figures 4 to 6 we target the following primary classification aerosol regimes: 1) the background regime, 2) the marine regime, and 3) the continental level 1 regime. These regimes are characterized by aerosol properties with much lower values than the other regimes (Fig. 2) and are therefore particularly important in preindustrial times and in the SSP1-1.9 scenario, where they cover most of the areas of the globe. The aim of the secondary classification is to zoom into these specific primary regimes to identify further differences and similarities within these regimes. We refer to the regimes characterized by the secondary classification as "sub-regimes" and aim to answer the following question "Where do preindustrial-like aerosol conditions most probably occur at present-day and in the different future scenarios?

The sub-regime distributions of the background regime (Fig. 4) reveals that preindustrial-like aerosol conditions over the polar regions in both hemispheres can be found also in the present-day case and that aerosol conditions of large parts of the continents in the Northern Hemisphere during pre-industrial times can still be found under present-day conditions, but only at very high northern latitudes (i.e., the border between continents and the Arctic ocean). The secondary classification identifies 5 sub-

regimes within the present-day background regime: one with enhanced sea salt over the oceans (sub-regime 0_0), one in the Arctic with the second lowest aerosol values (sub-regime 0_1), one in the Antarctic with lowest aerosol values (sub-regime 0_2), a dust-enhanced one at the southern edge of Australia (sub-regime 0_3), and one over the continental boundaries of the Northern Hemisphere with the highest aerosol values (sub-regime 0_4). During preindustrial times, the northern polar region

shows higher aerosol values than the south pole region, consistent with the present-day case. A possible explanation for this difference could be the influence of long-range transport of pollutants to the Arctic and new particle formation being favored under the very clean conditions over the Antarctic. The Antarctic sub-regime has the same extent in the preindustrial and in the present-day case, while the North Pole region sub-regime is larger than at present-day. Other continental regions belong to the sub-regime with highest aerosol values. The sub-regime distribution for SSP3-7.0 (Fig. 4f) closely resembles the present-

410 day conditions, while the results for SSP1-1.9 (Fig. 4d) are similar to the preindustrial PI-1850 patterns. SSP2-4.5 (Fig. 4e) shows an intermediate stage between SSP3-7.0 and SSP1-1.9.

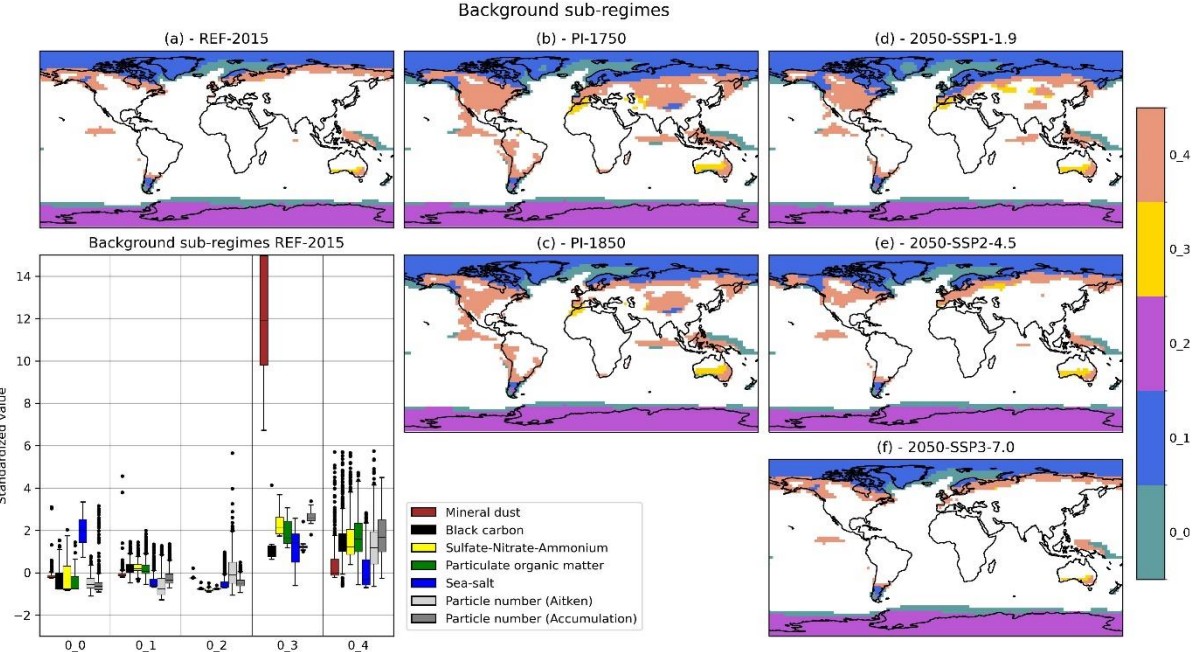

**Figure 4: Results of the secondary classification for the background regime. The sub-regime distribution is shown for present-day (a), preindustrial times (b-c) and future (d-f). The aerosol properties for each regime of the present-day case are shown on the bottom**

**left. Note that the standardized values here should not be directly compared with Fig. 3 and with other secondary classification results. The standardized value 0 of the secondary classifications represent the sample average of the target primary regime. The negative (positive) values describe data below (above) sample average expressed as factors of their sample standard deviation. Recall that the extent of the primary target regime (background) varies from case to case. Hence also the areas outside the target regime (white) vary.**

Over the oceans, preindustrial-like conditions can only be found in the Southern Hemisphere (Fig. 5), as also pointed out by several previous studies and measurement campaigns (e.g. Andreae, 2007; Carslaw et al., 2014; Hamilton et al., 2014; McCoy et al., 2020). The present-day marine regime is further classified into three sub-regimes. The secondary classification (Fig. 5c) further identifies the different aerosol sub-regimes between the northern hemispheric (sub-regime 4_2) and the southern

hemispheric oceans (sub-regime 4_0), with lower aerosol values in sub-regime 4_2 than 4_1. The marine regime contains a further sub-regime showing an enhancement in mineral dust, black carbon and POM at the outflow of the African continent (sub-regime 4_1) (Fig. 5d). This suggests that these sub-regimes are influenced by their neighbouring primary regimes to a certain extent, as a results of atmospheric long-range transport processes. In the preindustrial times and 2050 in SSP-1.9 scenario, the oceans are mostly classified into the sub-regimes 4_0, while only small regions in the tropics are identified as

sub-regime 4_2. The reasons could be that aerosol conditions over the continents of the Northern Hemisphere in preindustrial times were not polluted enough to be able to influence the northern hemispheric oceans in a sufficient degree which can be recognized by the applied classification rules. The sub-regime 4_1 at the African outflow is consistent with the present-day case. The distribution of sub-regimes of the marine regime is highly consistent between PI-1750 and PI-1850. The sub-regime distribution of SSP3-7.0 is almost identical with the present-day case, and SSP2-4.5 represents an intermediate stage between

preindustrial times and present-day.

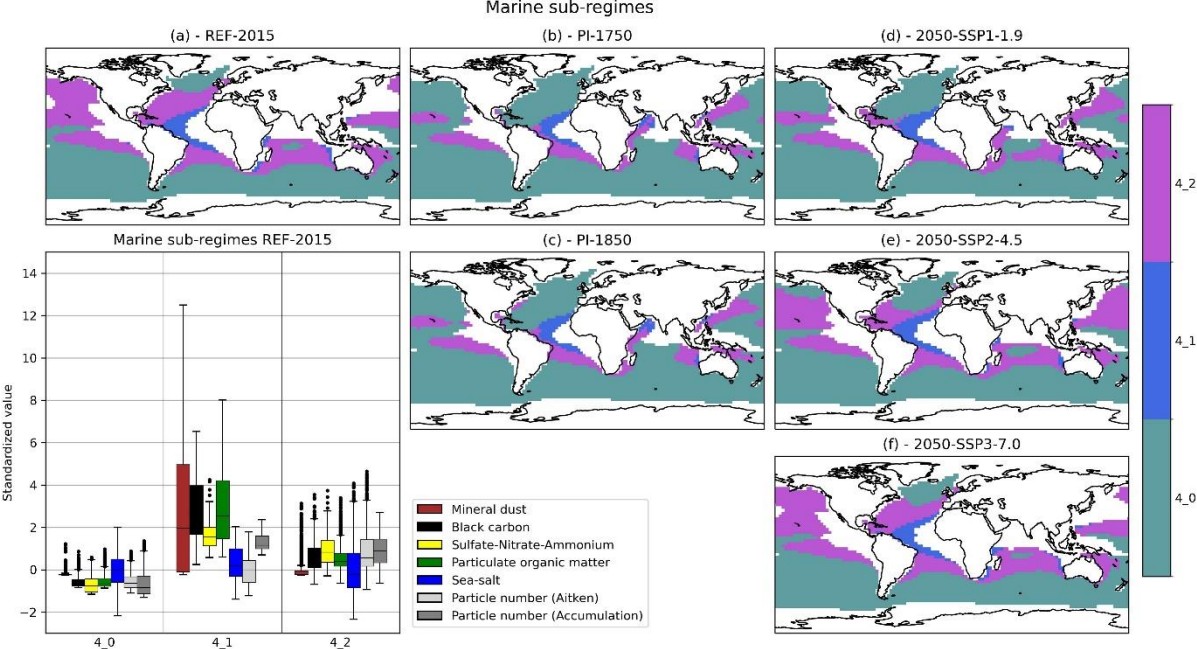

Figure 5: As in Fig. 4, but for the marine regime.

The sub-regimes of the continental level 1 regime show also locations were preindustrial-like conditions can be found in the present-day case (Fig. 6). The present-day continental level 1 regime contains a sub-regime 1_3 with dust enhancement, a sub-regime 1_2 with enhanced sea salt, a SNA-enhanced sub-regime 1_1 which is found at a latitudinal band across the oceans south of the Eurasian continent, a sub-regime 1_4 next to the primary biogenic regime, and a sub-regime 1_0 at northern regions. In the preindustrial times, the regime shows a similar sub-regime structure as in the present-day, but fine differences occur. The present-day sub-regime 1_1 is not found during the preindustrial times, it seems to be replaced by the cleaner sub-regime 1_2. During preindustrial times, the dust enhanced sub-regime 1_3 has a larger spatial extent than in the present-day case, the biogenic/biomass burning enhanced sub-regime 1_4 includes also southeast China and parts of India which are the most polluted regions in the present-day case. The sub-regime 1_0 during preindustrial times corresponds to present-day conditions in a few parts of North America and the Eurasian continent. Similar as the results for other secondary classification regimes, SSP3-7.0 closely resembles the present-day aerosol conditions, and SSP1-1.9 resembles PI-1850. SSP2-4.5 corresponds to an intermediate condition between preindustrial times and present-day.

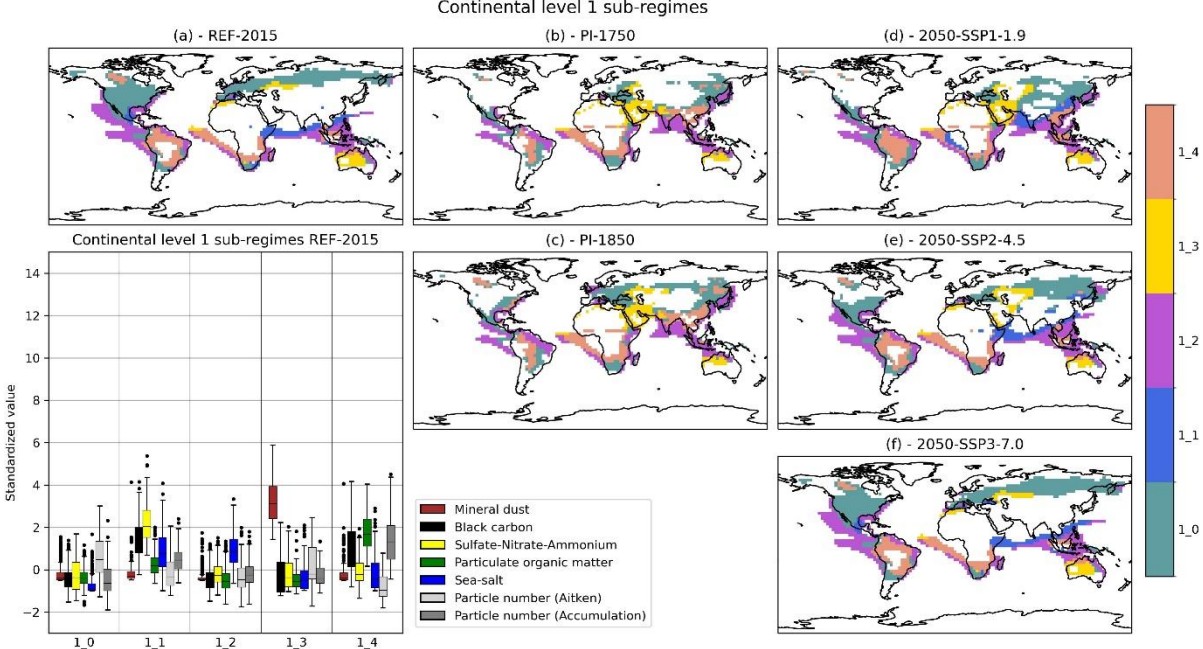

**Figure** 6**: As in Fig. 4, but for the continental level 1 regime.**

### 3.3 Regional focus and emission analysis

The goal of this section is to explain prominent features of the temporal development by means of analysing the development of the emissions in selected regions. The results discussed in the previous sections allowed us to identify six representative regions within three groups (biogenic, most polluted region and developed countries) (Fig. 7a). Each group contains two

regions which belong to the same aerosol regime during present-day, but follow different pathways in terms of their aerosol regime evolution from preindustrial times to the future. We investigate in this section which emission precursor gas / emission component contribute to the regime development of selected example regions. , by comparing the anthropogenic and biomass burning emissions of aerosol and precursor gases in the different time periods and scenarios for these regions (Fig. 7b-g). We consider three emission sectors (open burning, the sum of anthropogenic non-transport sectors, and the transport sector) and the sum of them. The transport sector includes land transport, shipping and aviation, with most of the emissions due to land transport. We consider four key species: $NH_3$ (mostly driven by agricultural activities), $SO_2$ and $NO_x$ (typical pollutants from fossil fuel combustion and both important for secondary aerosol formation), BC (representative for the emission from all kinds of combustion processes, including also open burning). Most of the emission-related anthropogenic aerosol changes will be addressed by the analysis of these species, which are representative for secondary aerosol formed from precursor gases and for primary aerosols. This type of analysis is possible because the simulations consider only emission-driven changes and neglect climate change effects on the analysed aerosol properties. The open burning sector comprises agricultural waste burning on fields, forest burning, grassland burning and peat burning (van Marle et al. 2017). The total emissions of each species are normalized with respect to their present-day total in each region, such that values of the total emissions larger (smaller) than 1 indicate an emission increase (decrease) with respect to the present-day.

Region R1a (over South America) and R1b (over Central Africa) are both identified as biogenic/biomass burning regimes during present-day. While R1b belongs to the biogenic/biomass burning regime at all time periods, R1a belongs to continental level 1 regime during preindustrial times and seems to develop into the biogenic/biomass burning regime only in the present-day case (Fig. 2). This behavior can be explained by the emission analysis (Fig. 7b, c). Preindustrial emissions for R1a are very low (less than ~1% of present-day), while total preindustrial emissions for R1b are about 50-90% of the present-day value (depending on the species). This dramatic change in R1a between preindustrial and present-day conditions points to a strong influence of human activities on the open burning emissions, e.g. due to deforestation. This also explains why the classification algorithm assigns R1a to a continental level 1 regime in the preindustrial time. In SSP1-1.9 the emissions in R1a amount to about 25-70% of the present-day value. The assignment of R1a to the continental level 1 regime under SSP1-1.9 suggests therefore that reducing ~50% of present-day emissions in R1a could reverse the direction of aerosol regime development in this region. R1a is barely influenced by open burning emissions in the preindustrial times, while this sector is a constant emission source in R1b over the different time periods.

The south-east Asian regions R2a and R2b develop into the most polluted aerosol regimes on Earth in the present-day case. Both regions belong to the continental level 1 regime in the preindustrial times but develop differently in terms of their future regimes (Fig. 2) which can be explained by their slightly different emission pathways in the future scenarios (Fig. 7d, e). The extremely low emissions during preindustrial times compared with the present-day result in large temporal differences in these regions. In 2050, R2a shows increased emissions of $NO_x$ and $NH_3$ under the SSP3-7.0 and SSP2-4.5 scenarios. In R2a $NO_x$

emissions increase by 30% for SSP2-4.5 and by 70% for SSP3-7.0 compared to the present-day case. An emission reduction can be seen for R2a under the clean scenario SSP1-1.9, with the exception of $NH_3$ which is strongly influenced by agricultural emissions. In R2b SSP3-7.0 still shows increases while SSP2-4.5 and SSP1-1.9 show decreasing emissions, except for $NH_3$,

again because of the persisting emissions from agriculture. The different pathways of emission changes in R2a and R2b can explain why R2a remains in the polluted regimes in 2050, while R2b shifts to a clean aerosol regime under SSP1-1.9. The transport sector has only small contributions for most species, except for $NO_x$. Compared to R2b, the relative contribution of the transport sector with respect to the total emissions is larger in R2a.

R3a and R3b are representative of developed countries and are covered by clean aerosol regimes. These regions show a gradual development from the background to the continental level 1 regime from past to present, and a reverse process from the present-day to the future (Fig. 2 and Fig. 6) under different degrees of emission reduction. By addressing the considered time periods, the emission comparisons for both regions (Fig. 7f and g) show that the emission maxima of $NO_x$, $SO_2$ and BC occur at present-day, while emissions for $NH_3$ increase up to 20% by 2050 under the most pessimistic SSP3-7.0 scenario. However, as we have

not analyzed the full time series, the maximum aerosol emissions could peak before or after present-day. This trend agrees with the temporal development of the corresponding aerosol regimes. The anthropogenic non-transport sector shows the largest contribution to the total emissions in these regions, followed by the contributions from the transport sector.

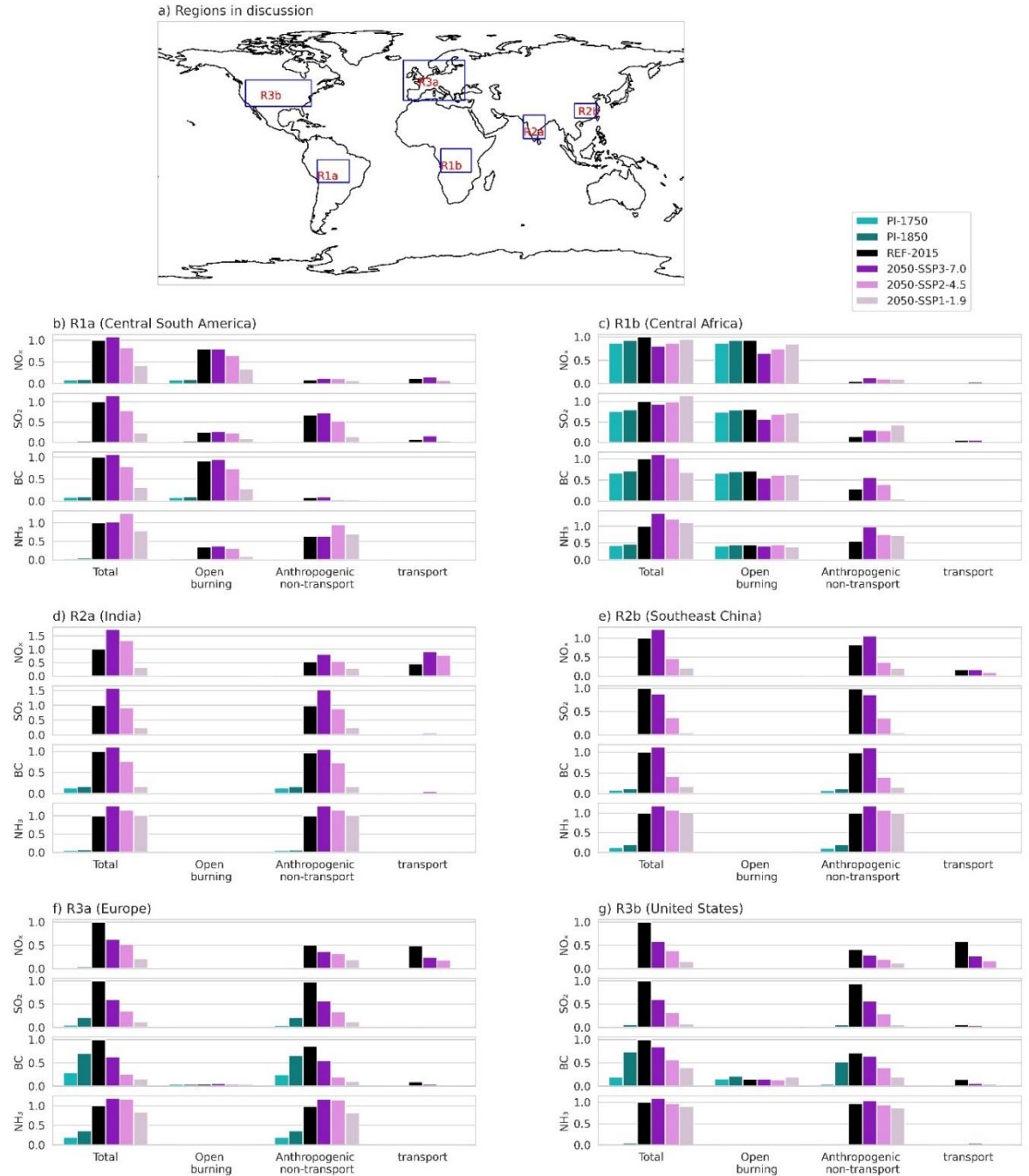

**Figure 7: a) shows the six regions selected for the emission analysis, and b)-g) show the comparison of emissions for four species and from different sectors during different time and scenarios for each region. The emission amounts of each species are normalized to their present-day total. Note the different vertical scales in panel (d).**

## 4 Limitations, strengths and potential applications

In order to set the results presented above in the right context we discuss the most important limitations, strengths and potential applications of our analysis in the following.

As already stated in L22, the classification algorithm is applied to the output of global model simulations and therefore cannot capture small-scale patterns (small-scale effects of individual pollution sources as, for instance, local plumes of specific roads, power plants, ships or oil platforms) due to the model's coarse resolution (~300 km in this study). Therefore, the presented results only consider large-scale patterns. In spite of this limitation, a clear advantage of using model-generated output is the large number of aerosol properties calculated in a self-consistent manner as well as the complete spatio-temporal coverage of the globe for multi-annual time periods. This, to some extent, compensates for the limitations mentioned above and the uncertainties of global aerosol models identified by comparing to observations. Of course, model simulations can be affected by biases visible as deviations from measurements. This, however, is less critical in the context of this study, due to the standardization process. The classification algorithm is based on assessing large-scale distribution patterns of aerosol properties and previous studies showed that these patterns are usually well captured by global models (Koch et al., 2009; Mann et al., 2014; Aquila et al., 2011; Koffi et al., 2015; Kaiser et al., 2019; Beer et al., 2020).

Another limitation specific to this study is related to the set-up of the model simulations, which considers time-varying emissions under fixed present-day climatic conditions. Although the intention of this study is to purely investigate the influence of emissions on the development of aerosol regimes over time, the simulations for the preindustrial times and the future may not be fully representative of their atmospheric conditions in the real world. Changing emissions and climate simultaneously is technically possible and would not affect the algorithm skills, but it would hamper the separation of their respective impacts, further complicating the interpretation of results and the applicability of the proposed method. The question is how this assumption could influence the results presented in this study. Previous studies show that climate change could affect dimethyl-sulfide (DMS) production (Bopp et al. 2003, Zhao et al. 2024), mineral dust (Kok et al. 2023), sea salt (Struthers, et al. 2013), $PM_{2.5}$ and aerosol optical depth (IPCC, 2022). These influences of climate change on natural emissions are not considered in our study, with the intention to clearly attribute the differences within the time slices and scenarios to the underlying emissions. Nevertheless, the influence of climate change on aerosols could be important and further studies are needed to investigate the relevance of this effect on the patterns of the identified aerosol regimes. However, previous studies suggested a stronger influence of emission changes on aerosols than climate change. Lacressonnière et al. (2016) investigated PM mass concentrations over Europe in a +2 °C warming world and demonstrated that the decrease of PM mass concentrations over Europe is mainly associated with emission reductions. Cholakian et al. (2019) investigated climatic drivers and their effect on PM10 components in Europe and the Mediterranean Sea and demonstrated that anthropogenic emission changes overshadow changes caused by climate for both regions. Li H. et al (2022) evaluated the contributions of emission changes and climate

change to the projection of $PM_{2.5}$ in 2100 and suggests that under clean emission scenarios (SSP1-2.6 and SSP2-4.5), the $PM_{2.5}$ reduction in 2100 is due to emission reductions, while for a high pollution scenario (SSP5-8.5) an approximately equal contribution of emission changes and climate change to $PM_{2.5}$ mass concentrations for specific world regions (e.g. South America, Asia) was identified. These studies support the validity of our conclusions drawn for pre-industrial times and under the two clean emission scenarios for 2050. However, our results for the regimes in 2050 under SSP3-7.0 may be subject to uncertainties due to neglected climate change effects, although here we focus on the year 2050, when the climate change effects in scenarios of high pollution are smaller than in 2100 (e.g. Fig SPM.8 in IPCC 2021 Summary for Policymakers), and the high emission scenario SSP3-7.0 which we address in this study is cleaner than SSP5-8.5 investigated by Li H. et al. (2022). In summary, the effect of climate change is suggested to be less important than the emission changes for the aerosol regimes investigated in our study. The missing climate change effects might still result in uncertainties in the size and extent of the regimes, but will likely not change their general patterns. Hence, the major conclusions of this study are unlikely to change when climate change is considered. The investigation of the influence of both climate and emission changes on the temporal development of aerosol regimes may nevertheless be the subject of a future study.

Another factor needs to be considered is that all our conclusions are driven by simulations using the adopted emission inventories. As stated above, these emission inventories might miss important emission phenomena, especially those occurring on small spatial and short temporal scales. However, on the climatological time scales analysed in this study, these limitations are not expected to change the conclusions. On the other hand, since we analyse long term mean conditions, the analysis cannot capture the fact that some areas may be subject to temporary pollution events (e.g., due to wild fires). Caution is required when using biomass-burning and biogenic emission datasets. A reliable representation of biomass-burning and biogenic emissions during pre-industrial times is not available (e.g. Marlon et al. 2016), and future climate-driven changes are unlikely to be properly represented in the CMIP6 emission inventory used to drive the simulations analysed here. This uncertainty might affect our conclusions regarding biomass-burning and biogenic regimes in terms of their size and extent during pre-industrial times and in the future.

This study uses an innovative way to assess and integrate information from multiple aerosol properties. Unlike the traditional single variable model assessments, which consider only one specific aerosol property for different time slices, we condense information from seven key aerosol properties into a single parameter (the regime index) and then assess the development of this parameter through time. In this way we identify regimes in the present-day lower troposphere with distinct characteristics (e.g., clean, dust-dominated, polluted, etc.). Moreover, using the present-day regime characteristics as a reference, we can compare the present-day case with other time slices to identify similarities and differences. If these comparisons among time slices were conducted for each aerosol property individually, the diverse and complex patterns for different aerosol properties would complicate the interpretation and make it more difficult to derive key information and draw general conclusions.

The analysis procedures developed in this study and the presented results are unique and have a large application potential. This combination of unsupervised and supervised machine learning methods is valuable for comparing consistent patterns across different datasets without knowing the a priori classification criteria. The application of this method is not limited to global aerosol simulations, but can be applied to different modelling and measurement (such as Satellite) datasets regardless of their resolution and spatial coverage. The clear and concise depiction of the spatial extent and distribution of aerosol regimes

for different time periods and emission scenarios is particularly useful. First, previous campaigns targeting preindustrial aerosol conditions took place mainly over the southern oceans. Our results point to additional areas with aerosol conditions similar to those at preindustrial times (such as the background regime, the biogenic regime in central Africa and the continental level 1 regime), which may be taken into consideration. Second, the representation of aerosol conditions for present-day and different future scenarios, derived from a complex dataset, provides insights for policy- and decision-makers on possible future

developments of atmospheric pollution. Third, the classification of aerosol conditions into regimes provides important hints for subsequent studies relying on information about the properties and distribution of atmospheric aerosols. For example, these regimes can be used as input for assessing the aging of aircraft engines dependent on atmospheric environmental conditions. Many aerosol components (e.g. sea salt, mineral dust, black carbon, sulphate) induce highly relevant engine aging processes, for instance through corrosion or abrasion (Ellis et al. 2021). Hence, detailed knowledge on aerosol characteristics in specific

regions is an important prerequisite for robust engine life cycle modelling. Our study addresses aerosol climatological conditions in a similar manner as the IPCC does when defining climate reference regions (Iturbide et al. 2020). Iturbide et al. (2020) defined 46 land and 16 ocean regions to represent consistent and climatically-coherent regional climate features based on observational and model-simulated temperature and precipitation data, in order to provide climatological information for a broad spectrum of IPCC users. Similarly, different communities (e.g., industry, policy, research) might benefit from the

aggerated aerosol information provided in our study.

## 5 Summary and outlook

This study investigated multiple aerosol properties simulated with the global aerosol model EMAC with the aerosol submodel MADE3 for different time periods, from preindustrial times to 2050 under three different emission scenarios (SSP3-7.0, SSP2-4.5 and SSP1-1.9). The simulations considered varying emission conditions for the different time periods, but the climate was

constrained by present-day reanalyses. The comparison of similarities and differences in aerosols properties over time was investigated by means of classifying aerosol regimes based on seven aerosol properties including aerosol mass concentrations of BC, mineral dust, sea salt, POM, the sum of sulfate, nitrate, and ammonium (SNA), as well as number concentrations of aerosol particles in the Aitken and accumulation modes. The analyses were based on two statistic machine learning algorithms (a combination of K-means and Random Forest). The investigations were conducted based on a primary classification of

aerosol regimes and a secondary classification of more detailed structures in selected regimes of the primary classification. The results led to the following new findings about aerosol patterns and their evolution:

(1)     The machine learning classification based on the present-day dataset (2015) (Fig. 2a and 3) identified eight aerosol regimes in the primary classification step: background, marine, biogenic/biomass burning, two dust-dominated regimes that we combined into one, and four continental regimes with different levels of aerosol burden (level 1 to level 4 characterized by increasing aerosol burden from level to level). During preindustrial times (years 1750 and 1850), clean aerosol regimes (background, marine, and continental level 1) dominate, except for mineral dust regimes in northern Africa and biomass burning/biogenic regimes in central southern Africa. The aerosol regimes for 2050 under SSP3-7.0 closely resemble present-day conditions, but there are some notable regional differences: developed countries tend to shift towards cleaner regimes in future, while the opposite is the case for developing countries. The aerosol regimes for 2050 under SSP2-4.5 and SSP1-1.9 develop towards cleaner aerosol conditions with respect to the present-day case, whereas SSP1-1.9 has the highest probability to resemble preindustrial conditions, more similar to PI-1850 than PI-1750. SSP2-4.5 mostly corresponds to an intermediate state between present-day and preindustrial times.

(2)     A secondary classification was conducted in order to characterize the clean aerosol regimes in more detail (Figs. 4-6). The results suggested that present-day northern polar regions show a higher level of aerosol pollution than the southern polar regions, and this tendency is also evident in the preindustrial times. Present-day marine regimes show a north/south contrast, while preindustrial marine regimes mostly represent cleaner marine conditions found over the present-day southern oceans.

(3)     An analysis of the emissions of black carbon and several aerosol precursor gases was performed targeting six example regions for three groups (biogenic, most polluted regions and developed countries) for different time periods. The example regions within a specific group belong to the same regime during present-day, but follow different pathways in terms of their evolution over time. The results suggested that i) the absence of the biogenic/biomass burning regime over South America during preindustrial times could result from preindustrial $NO_x$ and BC emissions accounting for less than ~10% and $NH_3$ and $SO_2$ accounting for less than ~1% of present-day emissions, although this conclusion is highly uncertain; ii) the behaviour of the most polluted present-day regime in India, showing a slower transition towards less polluted conditions than the corresponding regime in south-eastern China in 2050 for SSP1-1.9 and SSP2-4.5, could be due to different emission reduction policies; and iii) in developed countries, emission reductions could explain a shift towards preindustrial-like conditions in 2050 for all scenarios.

For the applicability of the methods discussed in this work, a few caveats and limitations need to be considered. First, the global aerosol simulations are capable of capturing large-scale aerosol distribution patterns, but are not able to resolve small-scale and localized processes. Therefore, the results consider solely large-scale patterns. Second, the atmospheric aerosol for preindustrial and future conditions is simulated by varying the emissions for the respective time period but with a fixed present-

day climate. Neglecting the effects of climate change is expected to cause uncertainties in the size and extent of the identified regimes (especially for polluted regimes under SSP3-7.0 scenario), but the major conclusions drawn in this study will likely be unaffected, due to a dominating influence of emissions than climate change as suggested by previous studies. The consideration of both climate and emission changes for the corresponding time periods could be the subject of a follow-up

study, which may also address the sensitivity of our results to the input model data considered in the clustering. Here we focused on simulation data from the EMAC model, but the same approach could be applied, for example, to the CMIP6 model output, although it may need to be adapted to the availability of the aerosol properties used to drive the machine learning algorithms.

The results presented in this study can provide important insights for different communities (e.g., industry, policy, research). For instance, our findings can be used for the planning of targeted measurements for specific aerosol conditions (e.g. preindustrial-like, most polluted, biogenic/biomass burning), for supplying information to policy makers for tailored air pollution mitigation measures in specific regions, for model inter-comparisons (e.g. AeroCom, Gliss et al. 2021; AerChemMIP, Collins et al. 2017), and also for applied studies, for instance, the life cycle modeling of aircraft engines, where information

on atmospheric aerosol properties are required as an essential input. The method we outlined here could also be applied to analyse the development of upper tropospheric aerosol regimes over time. This region is of particular interest for many areas of research, such as aviation climate impacts and volcanic emissions.

**Data availability**

The output of the model simulations investigated in this paper is available at https://doi.org/10.5281/zenodo.8134336 (Righi, 2023). The aggregated data generated in this study will be made available via doi in the final version of this paper.

**Author contributions**

JL conceived the study, developed and implemented the analyses procedures, analysed the data, and wrote the paper. MR

performed the simulations used in this study. JH and AS supervised this study. MR, JH, CGB, UB and AS contributed to conceiving the study, interpreting the results, and writing the text.

**Competing interests**

The contact author has declared that none of the authors has any competing interests.

**Acknowledgements**

We thank for Simon Kirschler (DLR, Germany) for his suggestions on an earlier version of this paper. We are grateful to the developer of the Python package scikit-learn for providing this excellent machine learning tool (https://scikit-learn.org/stable/).

The data analysis for this work used the resources of the Deutsches Klimarechenzentrum (DKRZ) granted by its Scientific Steering Committee (WLA) under project ID bd0080. Further, datasets provided by CMIP6 via the DKRZ data pool were used.

**Financial support**

This research has been supported by the DLR transport research projects TraK and DATAMOST, and the Bundesministerium für Wirtschaft und Klimaschutz (BMWK) (project: Digitally optimized Engineering for Services – DoEfS; contract no. 20X1701B).

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
