# Peer review of "Revealing dominant patterns of aerosols regimes in the lower troposphere and their evolution from preindustrial times to the future in global climate model simulations"

_EGUsphere, 2024_

## Referee Comment (RC1)

Comment to *"Revealing dominant patterns of aerosol regimes in the lower troposphere and their evolution from pre-industrial times to the future in global climate models"* by Jingmin Li et al.

This paper uses aerosol concentrations from simulations with one global model as input to machine learning algorithms in order to provide classification of aerosol regimes under pre-industrial, present-day and future emissions. The paper is mostly well-written and structured, and the underlying methodology sound. However, in the current version the statements and conclusions give very limited novel information and added value to the literature and scientific community, and I also question the usefulness of the method in the context of assessing the influence of projected anthropogenic emission changes. Improvements are needed to make the manuscript suitable for publication.

While the use of concept of aerosol regimes does provide a different view on future aerosols than AOD, PM2.5, individual species, etc (of which there have been several studies), the paper at the moment provides many statements, descriptions and conclusions that are well-known and shown in previous literature and/or simply follow directly from the underlying emissions that are used as input to the model. The classification method itself was also well documented in L22. Statements such as "suggesting a general reduction of aerosol and aerosol precursor emissions in line with the underlying assumptions in these scenarios." seems rather circular and given that there is no change in climate in these model runs, really only says that emissions have been properly read into the model and that the classification method works also for different emissions than those used in L22. Another example is "This trend agrees with the temporal development of the corresponding aerosol regimes." – when in fact this trend drives the corresponding aerosol regime change. The "emission analysis" also appears a bit simplistic. At the very least, revisions of the abstract, results and conclusions are needed to emphasize what's important/new knowledge from this paper, incl. e.g. comparing the usefulness of this ML approach and clustering with other model based assessments, and acknowledge how this work supports/complements other published work looking at future aerosol. There is some discussion of previous studies, but it cites only two rather old papers.

One question that arises is how sensitive the classifications, and the subsequent conclusions, are to the model input used? We know that global models have a widespread in simulated aerosol distributions and an extension of the is work that would bring added value is to consider multi-model data, e.g. from CMIP6. E.g. can robust regimes be identified for present-day and future scenarios? The authors should consider adding a multi-model perspective here, either for both present-day and future, or just present-day – or at the very least discuss this.

The classification is also a bit of a black box. While the point of the method is that the classification criteria is not known a priori, understanding the distinctions is important for further application. E.g. what is the criterium for calling something level 1 vs 4. This, combined with the naming, will in my view very much limit the relevance for policy-making/mitigation – which is one of the important applications the authors point out. It should be made clearer here (and at least to a reader who is not a ML expert) how the classification is done, e.g. what is the criteria for transitioning to a lower level continental airmass? What's the role of composition vs. mass concentration vs. number concentration – can level 1 continental air in fact be very differently composed in different time periods if emissions of different species change differently? If so, that would have subsequent implications for the climate effects, which would not be easily extracted. For policy relevance, could the different levels be related to e.g. air quality indices?

Another implication of the coarse aerosol regime approach is that a lot of detail is hidden. For instance, most of Africa is classified as dust-dominated and biogenic/biomass burning. In contrast, scenarios for anthropogenic emissions exhibit a wide range in future evolutions, which has significant implications for climate and air quality, but is not at all captured here. What is the authors' view on this? This draws into question the usefulness of such coarse classifications and I think the authors need to spend some more time justifying why their approach provides what they call a "clear and condensed" picture.

Another limitation is the fixed meteorology. The authors do talk about this, saying that "it would hamper the separation of their respective impacts, further complicating the interpretation of results and the applicability of the proposed method." In my view, this is where ML could be of added value, i.e. application of the algorithm to a more complex data set. While this is likely not possible here, a perhaps more important implication is that not having changing climate has implications for the aerosol load and composition in both future and pre-industrial times and requires more care than currently taken when talking about changes from the pre-industrial. For instance, studies have shown an increase in the dust loading and analysis of CMIP6 data have looked at possible feedbacks on natural aerosols. The authors should include a better discussion of such work and possible implications of these findings for their results.

The authors also discuss differences between 1750 and 1850: given the large uncertainties, I question the robustness of any conclusion drawn for this time period and the authors may want to acknowledge that more clearly.

Specific comments:

Section 3.2: why is transport singled out here from other non-transport emissions when this paper has no transport focus? In many cases there are little or no emissions and hence extracting only this sector does not help inform the changes.

Section 3.2: it would be helpful to have regions named rather than Rx.

Line 461: "transport sector shows the largest contribution to the total emissions in these regions, followed by the contributions from the transport sector" Well of course, you have not separated out any other anthropogenic sector… This statement gives no useful information.

Line 423: "Most of the processes driving the anthropogenic aerosol changes will be addressed by the analysis of these species" – what is meant by the word processes here? The reference is to the emissions documentation paper so I assume it's related to "processes" leading to emission changes – but could be misunderstood to involve also interactions between different species through atm. chemistry when emissions change (e.g. SOA formation changes when OC/POA emissions change). Moreover, this statement is probably true but that's because you have not change in climate, which should be specified. Perhaps rephrase.

Line 450: "The different pathways of emission changes in R2a and R2b can explain why R2a remains in the polluted regimes in 2050, while R2b shifts to a clean aerosol regime under SSP1-1.9" – can explain? What are the other possible explanations in this model study?

Line 457-459: "The emission comparisons for both regions (Fig. 7f and g) show that the emission maxima of NOx, SO2 and BC occur at present-day, while emissions for NH3 increase up to 20% in 2050 under the most pessimistic SSP3-7.0 scenario. However, the maximum aerosol emissions generally peak at present-day." Emissions in North America and Europe declined prior to 2015, so this is not accurate but appears to be the case because you don't show the full time series.

Line 479: "This, however, is less critical in the context of this study, due to the standardization process." I don't understand this statement. If you classify or standardize something that is not representative of the real world, how is that OK or not important?

Line 504: If referring to the dataset used in this study, then "huge and complex" seems a bit of an overstatement…

Line 506: given the list of co-authors I can see why aircraft engines are selected as an example, however, I struggle a bit with this example since the authors point. Given the coarse nature (in space and time) of the classification approach, how would the data be used in engine life cycle modeling?  And how would this better come from this study than all the other studies focusing on aerosol composition, with a full 3D spatial distribution?

---

## Author Comment (AC1)

Replies to Referee's Comments

We are grateful to the reviewers for their constructive comments, which helped us to improve the manuscript. A point-by-point reply to each comment can be found below: reviewer comments are shown in *italic*; our responses are shown as plain text and text passages from the manuscript are shown in blue. In the following, line numbers refer to the revised version of the manuscript with tracked changes.

**Response to Reviewer #1**
* * *
*This paper uses aerosol concentrations from simulations with one global model as input to machine learning algorithms in order to provide classification of aerosol regimes under preindustrial, present-day and future emissions. The paper is mostly well-written and structured, and the underlying methodology sound. However, in the current version the statements and conclusions give very limited novel information and added value to the literature and scientific community, and I also question the usefulness of the method in the context of assessing the influence of projected anthropogenic emission changes. Improvements are needed to make the manuscript suitable for publication.*

**Response**: We thank the reviewer for their many careful remarks on our study. Please find below our replies to the issues raised.

*While the use of concept of aerosol regimes does provide a different view on future aerosols than AOD, PM2.5, individual species, etc (of which there have been several studies), the paper at the moment provides many statements, descriptions and conclusions that are well-known and shown in previous literature and/or simply follow directly from the underlying emissions that are used as input to the model. The classification method itself was also well documented in L22. Statements such as "suggesting a general reduction of aerosol and aerosol precursor emissions in line with the underlying assumptions in these scenarios." seems rather circular and given that there is no change in climate in these model runs, really only says that emissions have been properly read into the model and that the classification method works also for different emissions than those used in L22. Another example is "This trend agrees with the temporal development of the corresponding aerosol regimes." – when in fact this trend drives the corresponding aerosol regime change. The "emission analysis" also appears a bit simplistic. At the very least, revisions of the abstract, results and conclusions are needed to emphasize what's important/new knowledge from this paper, incl. e.g. comparing the usefulness of this ML approach and clustering with other model based assessments, and acknowledge how this work supports/complements other published work looking at future aerosol. There is some discussion of previous studies, but it cites only two rather old papers.*

**Response (major comment #1)**: We agree that some statements in the manuscript could be misinterpreted. The intention of this study is to identify dominant patterns of aerosol regimes based on global model simulations of aerosol properties for different time slices. The emission analysis aims to link the contribution of specific emission components and sectors to the aerosol changes for selected representative cases. However, we disagree with the

reviewer, that the conclusions of our study can be drawn by analyzing the emission data only because our aerosol simulations include representations of the aerosol number concentration, size distribution and composition resolving several different aerosol modes, based on detailed three-dimensional simulations of the chemistry of aerosol precursor gases, aerosols microphysics and long-range transport (as explained in the Methodology Section 2.1). This is of high relevance since such kind of data is required to properly describe the details of the global aerosol population and this can only be provided by a global model simulating all the relevant processes. The clustering is based on multiple simulated aerosol properties rather than just the emission of primary aerosols and aerosol precursor gases provided by the emission inventories: the emitted precursors undergo different processes and transformations in the model to form secondary aerosol which, in addition to primary aerosol, is our subject of interest, rather than the precursors. Hence, this work goes beyond the simple analysis of emissions. We have addressed the above comments accordingly and added corresponding clarifications to the manuscript:

- The abstract has been revised according to the reviewer´s suggestions as follows: "Aerosols play an important role in the Earth system, but their impact on cloud properties and the resulting radiative forcing of climate remains highly uncertain. The large temporal and spatial variability of a number of aerosols properties and the choice of different 'pre-industrial' reference years prevent a concise understanding of  their impacts on clouds and radiation. In this study, we characterize the spatial patterns and long-term evolution of lower tropospheric aerosols (in terms of regimes) by clustering multiple instead of single aerosol properties from preindustrial times to the year 2050 under three different Shared Socioeconomic Pathway (SSP) scenarios. The clustering is based on a combination of statistic-based machine learning algorithms and output from emissions-driven global aerosol model simulations, which do not consider the effects of climate change . Our analysis suggests that in comparison with the present-day case, lower tropospheric aerosol regimes during preindustrial times are mostly represented by regimes of comparatively clean conditions whereby marked differences between the years 1750 and 1850 emerge due to the growing influence of agriculture and other anthropogenic activities in 1850. Key aspects of the spatial distribution and extent of the aerosol regimes identified in year 2050 differ compared to pre-industrial and present-day, with significant variations resulting from the emission scenario investigated. In 2050, the low emission SSP1-1.9 scenario is the only scenario where the spatial distribution and extent of the aerosol regimes very closely resembles preindustrial conditions whereby the similarity is greater compared to 1850 than 1750. The aerosol regimes for 2050 under SSP3-7.0 closely resemble present-day conditions, but there are some notable regional differences: developed countries tend to shift towards cleaner conditions in future, while the opposite is the case for developing countries. The aerosol regimes for 2050 under SSP2-4.5 represent an intermediate stage between preindustrial times and present-day. Further analysis indicates a north/south difference in the clean background regime during preindustrial times, and close resemblance of pre-industrial aerosol conditions in the marine regime to present-day conditions in the Southern Hemispheric ocean. Not considering the effects of climate change is expected to cause uncertainties in the size and extent of the identified aerosol regimes but not the general regime patterns, due to a dominating influence of emissions rather than climate change in most cases.  The approach and

findings of this study can be used for designing targeted measurements of different preindustrial-like conditions, and for tailored air pollution mitigation measures in specific regions.".

- To address the issue of climate-change-induced vs. emission-driven changes, a brief discussion of the study by Li H. et al. (2022, not to be confused with our previous publication L22) on the future PM2.5 mass concentrations has been included in the introduction (line 65): "Li H. et al. (2022) estimated future PM2.5 mass concentrations (aggregated mass of particles less than 2.5 µm in diameter) by applying a Random Forest regression method to global atmospheric chemistry model results and CMIP6 multi-model climate projections. Their study suggests that under low- and medium-emission scenarios (SSP1-2.6 and SSP2-4.5) PM2.5 mass concentration decreases by about 40% in East Asia, 20−35% in South Asia, and 15−25% in Europe and North America in 2100 compared to present-day, and that the changes are mainly due to the emission reductions. Only in a high-end radiative forcing scenario (SSP 5-8.5), there is a comparable contribution of changes in climate and emissions to future PM2.5 changes over many regions on the Earth (e.g. East Asia, South Asia, Europe and North America)." Additional discussions on climate change effects have also been included in the manuscript (see our response to the major comment #5 below regarding the limitation of the fixed meteorology).
- The conclusions are modified similarly as the abstract: please refer to the changes in the Summary and Outlook section of the revised manuscript.

We further elaborate on the issue of the emission analysis below, when addressing the corresponding specific comments.

*One question that arises is how sensitive the classifications, and the subsequent conclusions, are to the model input used? We know that global models have a widespread in simulated aerosol distributions and an extension of the is work that would bring added value is to consider multi-model data, e.g. from CMIP6. E.g. can robust regimes be identified for present-day and future scenarios? The authors should consider adding a multi-model perspective here, either for both present-day and future, or just present-day – or at the very least discuss this.*

**Response (major comment #2)**: We understand the point about the sensitivity of our method to the input model data used for the clustering and appreciate the suggestion about the CMIP6 multi-model data, but we consider this kind of analysis beyond the scope of this study because our current analysis makes use of 7 different aerosol properties from the EMAC model output and not all of these properties are available from CMIP6 model simulations. This is the case, for instance, for the particle number concentrations and the particle size distributions. As shown in Figs. 3 and 4 in the manuscript (pasted below for convenience), several of the identified regimes are controlled and determined by particle number concentrations, which allows us to distinguish different regimes that are dominated by the same aerosol types but differ in terms of their magnitudes. For example, this allows us to distinguish between four different Continental regimes (Fig. 3), and between Arctic and Antarctic background sub-regimes using the secondary classification shown in Fig. 4.

Another advantage of using the EMAC output is our long-term experience with developing and evaluating this model, which facilitates the interpretation of the results, as we know in detail how the aerosol processes are represented in this model. Using CMIP6 multi-model output would make such interpretation much more challenging. Finally, we note that the simulations used here were performed in nudged mode, which provides a more accurate

representation of the meteorology compared to CMIP6-type free-running climate simulations, at least for present-day conditions. This is very important for the representation of specific aerosol properties, especially for the wind-driven aerosol species like mineral dust and sea-salt, which are relevant for the identification of specific regimes. Nevertheless, an extension of the current study to a different set of model data is certainly an interesting aspect and we suggest this as a future step. We revised the Summary and Outlook Sections as follows (line 650): "Neglecting the effects of climate change is expected to cause uncertainties in the size and extent of the identified regimes (especially for polluted regimes under SSP3-7.0 scenario), but the major conclusions drawn in this study will likely be unaffected, due to a dominating influence of emissions than climate change as suggested by previous studies. The consideration of both climate and emission changes for the corresponding time periods could be the subject of a follow-up study, which may also address the sensitivity of our results to the input model data considered in the clustering. Here we focused on simulation data from the EMAC model, but the same approach could be applied, for example, to the CMIP6 model output, although it may need to be adapted to the availability of the aerosol properties used to drive the machine learning algorithms.".

[Figure]

Fig. 3 from the manuscript: Internal aerosol properties of the primary classification aerosol regimes.

[Figure]

Fig. 4 from the manuscript: Results of the secondary classification for the background regime.

*The classification is also a bit of a black box. While the point of the method is that the classification criteria is not known a priori, understanding the distinctions is important for further application. E.g. what is the criterium for calling something level 1 vs 4. This, combined with the naming, will in my view very much limit the relevance for policy-making/mitigation –which is one of the important applications the authors point out. It should be made clearer here (and at least to a reader who is not a ML expert) how the classification is done, e.g. what is the criteria for transitioning to a lower level continental airmass? What's the role of composition vs. mass concentration vs. number concentration – can level 1 continental air in fact be very differently composed in different time periods if emissions of different species change differently? If so, that would have subsequent implications for the climate effects, which would not be easily extracted. For policy relevance, could the different levels be related to e.g. air quality indices.*

**Response (major comment #3)**: Fig. 3 in the manuscript (also shown above for convenience) is the key to answer these questions. Understanding the standardization process is the key to understand Fig. 3.

Unlike the more-complex deep learning algorithm, like for instance the neural networks, the statistics-based machine learning algorithms (e.g. K-means and Random Forest) applied here are relatively simple and it is possible to reconstruct what these algorithms are doing. K-means performs the classification based on the variances of a dataset, i.e. it divides the dataset into K (number of clusters) equal-variances clusters based on the implemented equal-variance classification criterion. This results in the distinct characteristics of the identified regimes shown in the box plot of Fig. 3. This box plots contains indeed the detailed information on composition, mass concentration, Aitken- and accumulation-mode number concentration of each regime, as the reviewer is asking.

Keeping in mind that the data are standardized, the highest standardized value for a specific aerosol parameter represents its global maximum and the lowest standardized value represents

its global minimum, because the standardization process normalizes values of different aerosol properties to the same order of magnitude, while conserving the underlying distribution of these aerosol properties. The different colors represent different aerosol properties (see the legend). Their values in specific regimes are represented in the box plot style, that is, a higher standardized value corresponds to a higher mass or number concentration in that regime. The naming of the regimes is admittedly subjective and needs to be interpreted together with Fig. 3. The term 'level' corresponds to different magnitudes of aerosol mass and number concentrations, ranging from low to high. Mapping our levels of the continental regimes to air quality indices would be a worthwhile future attempt to make our data directly usable by policy makers. In our previous study (L22), which addressed only present-day conditions, the terms 'slightly polluted' or 'moderately polluted' were used instead of the term 'level'. But in the present study we also investigate preindustrial conditions, for which the term 'polluted' does not sound appropriate. Therefore, we changed the naming to 'level' to define different levels of aerosol mass and number concentrations.

To address the points raised by the reviewer, the following text has been added to the manuscript when introducing Fig. 3 at line 289: "Fig. 3 shows comprehensive and integrated information on the present-day regime characteristics in terms of individual aerosol properties as classified by means of by K-means. In our approach, these K-means classification results also serve as a learning criterion for the Random Forest classification for other time periods. The different colors in Fig. 3 represent the different aerosol properties considered in this study. The y-axis shows their standardized values, with a higher (lower) standardized value corresponding to a higher (lower) aerosol mass and number concentration (i.e. the standardization process normalizes values of different aerosol properties to the same order of magnitude, while conserving the underlying distribution of these aerosol properties). The standardized values are also used to define the different pollution levels in the continental and dust-dominated clusters. The Random Forest algorithm learns from value ranges and the relative importance of the considered aerosol properties for each regime (regime characteristic), and then maps the pre-industrial and 2050 aerosol properties to the identified regimes. The same regime identified during pre-industrial times and 2050 represents the same conditions as the present-day regime (evaluated in the Fig. S1). We recall that the simulations analyzed here only consider the impact of changing emissions, while the impact of climate change is neglected. This might affect the size and extent of pre-industrial and future regimes to a certain extent but it should not change the classification substantially, since previous studies suggested a distinctively larger importance of emission changes than climate change for the evolution of the lower tropospheric aerosol (see detailed discussions in Sect. 4).".

*Another implication of the coarse aerosol regime approach is that a lot of detail is hidden. For instance, most of Africa is classified as dust-dominated and biogenic/biomass burning. In contrast, scenarios for anthropogenic emissions exhibit a wide range in future evolutions, which has significant implications for climate and air quality, but is not at all captured here. What is the authors' view on this? This draws into question the usefulness of such coarse classifications and I think the authors need to spend some more time justifying why their approach provides what they call a "clear and condensed" picture.*

**Response (major comment #4)**: As explained in our response to the previous comments, we use an integrated way to visualize information on the globally most-prominent aerosol characteristics. More details within the respective regimes can be identified by means of the secondary classification. Some examples of this are provided in the manuscript.

We agree that this could have been explained better in the original manuscript and we have added further explanations to clarify this. An additional paragraph in Sect. 4 (line 575) explains our statement about the 'clear and condensed picture': "This study uses an innovative way to assess and integrate information from multiple aerosol properties. Unlike the traditional single variable model assessments, which consider only one specific aerosol property for different time slices, we condense information from seven key aerosol properties into a single parameter (the regime index) and then assess the development of this parameter through time. In this way we identify regimes in the present-day lower troposphere with distinct characteristics (e.g., clean, dust-dominated, polluted, etc.). Moreover, using the present-day regime characteristics as a reference, we can compare the present-day case with other time slices to identify similarities and differences. If these comparisons among time slices were conducted for each aerosol property individually, the diverse and complex patterns for different aerosol properties would complicate the interpretation and make it more difficult to derive key information and draw general conclusions.".

*Another limitation is the fixed meteorology. The authors do talk about this, saying that "it would hamper the separation of their respective impacts, further complicating the interpretation of results and the applicability of the proposed method." In my view, this is where ML could be of added value, i.e. application of the algorithm to a more complex data set. While this is likely not possible here, a perhaps more important implication is that not having changing climate has implications for the aerosol load and composition in both future and pre-industrial times and requires more care than currently taken when talking about changes from the pre-industrial. For instance, studies have shown an increase in the dust loading and analysis of CMIP6 data have looked at possible feedbacks on natural aerosols. The authors should include a better discussion of such work and possible implications of these findings for their results.*

**Response (major comment #5)**: We thank the referee for bringing this up and agree that this has not been sufficiently discussed in the original manuscript. We have included a detailed discussion about the possible influence of meteorology and climate change on aerosols (line 538) as follows: "The question is how this assumption could influence the results presented in this study. Previous studies show that climate change could affect dimethyl-sulfide (DMS) production (Bopp et al. 2003, Zhao et al. 2024), mineral dust (Kok et al. 2023), sea salt (Struthers, et al. 2013), $PM2.5$ and aerosol optical depth (IPCC, 2022). These influences of climate change on natural emissions are not considered in our study, with the intention to clearly attribute the differences within the time slices and scenarios to the underlying emissions. Nevertheless, the influence of climate change on aerosols could be important and further studies are needed to investigate the relevance of this effect on the patterns of the identified aerosol regimes. However, previous studies suggested a stronger influence of emission changes on aerosols than climate change. Lacressonnière et al. (2016) investigated PM mass concentrations over Europe in a +2 °C warming world and demonstrated that the decrease of PM mass concentrations over Europe is mainly associated with emission reductions. Cholakian et al. (2019) investigated climatic drivers and their effect on $PM_{10}$ components in Europe and the Mediterranean Sea and demonstrated that anthropogenic emission changes overshadow changes caused by climate for both regions. Li H. et al (2022) evaluated the contributions of emission changes and climate change to the projection of $PM_{2.5}$ in 2100 and suggests that under clean emission scenarios (SSP1-2.6 and SSP2-4.5), the $PM_{2.5}$ reduction in 2100 is due to emission reductions, while for a high pollution scenario (SSP5-8.5) an approximately equal contribution of emission changes and climate change to $PM_{2.5}$ mass concentrations for specific world regions (e.g. South America, Asia) was identified. These studies support the validity of our conclusions drawn for pre-industrial times and under

the two clean emission scenarios for 2050. However, our results for the regimes in 2050 under SSP3-7.0 may be subject to uncertainties due to neglected climate change effects, although here we focus on the year 2050, when the climate change effects in scenarios of high pollution are smaller than in 2100 (e.g. Fig SPM.8 in IPCC 2021 Summary for Policymakers), and the high emission scenario SSP3-7.0 which we address in this study is cleaner than SSP5-8.5 investigated by Li H. et al. (2022). In summary, the effect of climate change is suggested to be less important than the emission changes for the aerosol regimes investigated in our study. The missing climate change effects might still result in uncertainties in the size and extent of the regimes, but will likely not change their general patterns. Hence, the major conclusions of this study are unlikely to change when climate change is considered.".

*The authors also discuss differences between 1750 and 1850: given the large uncertainties, I question the robustness of any conclusion drawn for this time period and the authors may want to acknowledge that more clearly.*

**Response (major comment #6)**: The debate whether using 1750 or 1850 as a reference for pre-industrial times is ongoing in the literature, as we noted in the Introduction (line 45). Our study highlights that there are likely differences in aerosol conditions between 1750 and 1850. We used CMIP6 emission inventory to drive our simulations, since it is a well-established and widely used dataset.

*Specific comments:*

*Section 3.2: why is transport singled out here from other non-transport emissions when this paper has no transport focus? In many cases there are little or no emissions and hence extracting only this sector does not help inform the changes.*

**Response**: Indeed, the paper has no transport focus, but the simulations on which we based the analysis stem from an assessment of the global impact of the emissions of the transport sector on aerosol and climate (Righi et al., 2023) and one motivation for our study is to develop a method, which could later be applied to analyze regions where transport emissions could have a large impact. This is because the emission patterns of transport are peculiar with respect to the other anthropogenic sectors: shipping, for instance, is the only anthropogenic source over the ocean and looking at this sector can help to interpret the marine regimes. Even an information about little or no emissions from transport sectors could be useful to interpret the changes across the different time slices.

*Section 3.2: it would be helpful to have regions named rather than Rx.*

**Response**: We have included regions names in Fig. 7 for more clarity, but we would prefer to keep the names R1a R1b, R2a R2b, R3a R3b because they specify three distinct groups, each containing two different regions a and b which are discussed together in the text.

*Line 461: "transport sector shows the largest contribution to the total emissions in these regions, followed by the contributions from the transport sector" Well of course, you have not separated out any other anthropogenic sector… This statement gives no useful information.*

**Response**: See our reply to a similar comment above, about our motivation to separate transport from the other anthropogenic sectors.

*Line 423: "Most of the processes driving the anthropogenic aerosol changes will be addressed by the analysis of these species" – what is meant by the word processes here? The reference is to the emissions documentation paper so I assume it's related to "processes" leading to emission changes – but could be misunderstood to involve also interactions between different species through atm. chemistry when emissions change (e.g. SOA formation changes when OC/POA emissions change). Moreover, this statement is probably true but that's because you have not change in climate, which should be specified. Perhaps rephrase.*

**Response**: This is indeed a confusing sentence. We thank the reviewer for pointing this out. The sentence (line 469) is modified to: "Most of the emission-related anthropogenic aerosol changes will be addressed by the analysis of these species, which are representative for secondary aerosol formed from precursor gases and for primary aerosols. This type of analysis is possible because the simulations consider only emission-driven changes and neglect climate change effects on the analysed aerosol properties.".

*Line 450: "The different pathways of emission changes in R2a and R2b can explain why R2a remains in the polluted regimes in 2050, while R2b shifts to a clean aerosol regime under SSP1-1.9" – can explain? What are the other possible explanations in this model study?*

**Response**: This sentence simply summarizes the considerations expressed in the previous sentences: since we neglect the effect of climate change, we can clearly attribute the differences within the three scenario cases to the underlying emissions. If we had considered both changes in climate and emissions, the interpretation of this results would be a challenge.

*Line 457-459: "The emission comparisons for both regions (Fig. 7f and g) show that the emission maxima of NOx, SO2 and BC occur at present-day, while emissions for NH3 increase up to 20% in 2050 under the most pessimistic SSP3-7.0 scenario. However, the maximum aerosol emissions generally peak at present-day." Emissions in North America and Europe declined prior to 2015, so this is not accurate but appears to be the case because you don't show the full time series.*

**Response**: Correct, thanks for pointing this out. We have modified these sentences (line 505) to: "By addressing the considered time periods, the emission comparisons for both regions (Fig. 7f and g) show that the emission maxima of NOx, SO2 and BC occur at present-day, while emissions for NH3 increase up to 20% by 2050 under the most pessimistic SSP3-7.0 scenario. However, as we have not analyzed the full time series, the maximum aerosol emissions could peak before or after present-day.".

*Line 479: "This, however, is less critical in the context of this study, due to the standardization process." I don't understand this statement. If you classify or standardize something that is not representative of the real world, how is that OK or not important?*

**Response**: The standardization changes the values of the input data but preserves the underlying geographical distributions of aerosol properties, which is what matters for the applied machine learning algorithms. The role of standardization has been systematically investigated and explained in our previous study (L22, see Section 4.1): basically, the standardization process removes the differences in value magnitudes and units from different aerosol properties while preserving the distributions of these aerosol properties, as shown by this figure from L22 comparing the probability density functions for several aerosol properties before and after standardization:

[Figure]

Probability density functions (PDFs) of selected aerosol properties (rows) derived from their global lower-tropospheric distributions in the raw (unscaled) data (left column) and after applying the standardization (right column). Adapted from Fig. 7 in L22.

With regard to the distribution of aerosol properties, it is stated in the manuscript (line 529) that "The classification algorithm is based on assessing large-scale distribution patterns of aerosol properties and previous studies showed that these patterns are usually well captured by global models (Koch et al., 2009; Mann et al., 2014; Aquila et al., 2011; Koffi et al., 2015; Kaiser et al., 2019; Beer et al., 2020).".

*Line 504: If referring to the dataset used in this study, then "huge and complex" seems a bit of an overstatement…*

**Response**: We agree with this comment and deleted 'huge' from the manuscript. However, this method can also be applied to even larger and more complex datasets.

*Line 506: given the list of co-authors I can see why aircraft engines are selected as an example, however, I struggle a bit with this example since the authors point. Given the coarse nature (in space and time) of the classification approach, how would the data be used in engine life cycle modeling? And how would this better come from this study than all the other studies focusing on aerosol composition, with a full 3D spatial distribution?*

**Response**: This example was inspired by a recent collaboration with an aviation industry partner, where we provided a dataset based on the condensed analyses of aerosol regimes, as an alternative to the full output from the global model simulations. The clustered dataset has proven useful to the aviation industry partner for pre-classification of engine damage potential in specific regions and engine lifecycle modelling.

**Response to Reviewer #2**

\*\*\*\*\*\*\*\*\*\*\*\*\*\*\*\*\*\*\*\*\*\*\*\*\*\*\*\*\*\*\*\*\*\*\*\*\*\*\*\*\*\*\*\*\*

*In this study, the authors improve and apply an aerosol regime analysis published previously (Li et al. 2022, L22) to past and future simulations of aerosols in the ECHAM-MESSY climate model. They find that cleaner regimes dominated in 1750 and 1850 and may dominate again in the future depending on the emission scenario.*

**Response**: We appreciate the supportive comments and constructive suggestions from the reviewer. Please find our replies to the issues raised below.

*The paper is well written, and figures illustrate the discussion well. However, the study suffers from only quantifying the emission-driven changes in aerosol regimes, not looking at climate-driven changes. This limitation has two consequences that severely reduces the insights gained by the study:*

- *First, there is little point in performing a regime analysis, or even in running a climate model. All the information is already in the emissions. This is particularly obvious in section 3.2, which reaches the same conclusions as the preceding sections but based on emissions only.*
- *Second, focusing on emissions leads to the probably very misleading conclusion that some future scenarios lead back to preindustrial conditions. That is unlikely: oceans will have warmed, soils and forests will have changed, bare soils will have expanded or shrunk, droughts and other weather extremes will be more frequent. All those changes will affect emissions from land- and ocean-based biogenic aerosols, biomass-burning aerosols, mineral dust, thus affecting the aerosol regimes, as noted in L22 and lines 136.*

**Response (Major comment #1)**: Thank you for this comment. Regarding the first point, we do not agree with the statement that the regime analysis could have been done by only using the emission data. Aerosol-chemistry model simulations provide much more insight into the details of the global aerosol population than analyses of emissions of aerosol precursor gases and primary aerosol species alone. Our aerosol simulations include representations of the aerosol number concentration, size distribution and composition resolving several different aerosol modes, based on detailed three-dimensional calculations of the chemistry of aerosol precursor gases, aerosol microphysics and long-range transport (as explained in the Methodology Section 2.1). Despite being driven by emissions of primary aerosols and aerosol precursors, the resulting atmospheric aerosol concentrations are strongly influenced by atmospheric transport and by chemical and microphysical processes. The emitted precursors undergo different processes and transformations in the model to form secondary aerosols. In addition, the simulations provide highly-relevant information on the number concentration, composition, and size distributions of different aerosol modes, which is not provided by emission data, but is essential for evaluating aerosol effects on climate and air quality. Our clustering method is based on multiple simulated aerosol properties rather than just the emission of primary aerosols and aerosol precursor gases. For the above reasons, this cannot be achieved by analyzing emission data alone. Hence, this work goes beyond the simple analysis of emissions. The motivation of the emission analyses performed in the manuscript is to explain which emission sectors most likely contribute to the changes in aerosols for the given example regions, but cannot be a replacement of the detailed global model experiments.

For the second point, we thank the referee for bringing this up and agree that this has not been sufficiently discussed in the original manuscript. We have included a detailed discussion on the possible impact of the missing influences of climate change on our results, and motivated the validity of our results under this assumption. Please refer to our reply to the major comment #5 of Reviewer 1, who raised a similar concern. We have also modified the Abstract and Summary and Outlook section to include this aspect in this study (see our reply to major comment #1 of Reviewer 1).

*Still, there are insights to be gained from an emission-only focus, which is why I recommend major revisions. A possible way to address the criticisms above could be to:*

- *Connect changes in emissions in a given region to changes in aerosol regimes over a wider area. That would involve merging section 3.2 with the rest of the analysis.*
- *Clearly acknowledge and flag the key limitation of the study in the abstract and add a long discussion to section 4, to elaborate on how climate feedbacks might alter the conclusions. There is an increasing body of work on climate-driven aerosol feedbacks. Chapters 6 (and perhaps 7) of the IPCC AR6 is a good starting point.*

**Response (Major comment #2)**: We are grateful to the reviewer for these suggestions. Regarding the first point on connecting changes in emissions to changes in aerosol regimes over a wider area, we discussed this possibility but we think it is not feasible here. Depending on the considered time period, a specific aerosol regime appears in different locations and its size and extent may vary. Some regimes are missing in certain time periods, e.g., the most polluted continental regime in SSP2-4.5 and SSP3-7.0 in 2050 is missing in the pre-industrial cases. Moreover, we think that presenting both regime and emission analyses together might overload this section and impair the readability. Hence, we prefer to first present the aerosol regimes and the emission analysis afterwards in a dedicated section.

Regarding the second point, we modified the abstract to better highlight the motivation and the limitations of our study (see our reply to major comment #1 of Reviewer 1). We also added a brief review of the relevant literature on the topic of climate- vs. emission-driven changes and included a detailed discussion of the possible impact of climate change on our analysis in Sect. 4, also in reply to a similar comment by Reviewer 1: "The question is how this assumption could influence the results presented in this study. Previous studies show that climate change could affect dimethyl-sulfide (DMS) production (Bopp et al. 2003, Zhao et al. 2024), mineral dust (Kok et al. 2023), sea salt (Struthers, et al. 2013), PM2.5 and aerosol optical depth (IPCC, 2022). These influences of climate change on natural emissions are not considered in our study, with the intention to clearly attribute the differences within the time slices and scenarios to the underlying emissions. Nevertheless, the influence of climate change on aerosols could be important and further studies are needed to investigate the relevance of this effect on the patterns of the identified aerosol regimes. However, previous studies suggested a stronger influence of emission changes on aerosols than climate change. Lacressonnière et al. (2016) investigated PM mass concentrations over Europe in a +2 °C warming world and demonstrated that the decrease of PM mass concentrations over Europe is mainly associated with emission reductions. Cholakian et al. (2019) investigated climatic drivers and their effect on $PM_{10}$ components in Europe and the Mediterranean Sea and demonstrated that anthropogenic emission changes overshadow changes caused by climate for both regions. Li H. et al (2022) evaluated the contributions of emission changes and climate change to the projection of $PM_{2.5}$ in 2100 and suggests that under clean emission scenarios (SSP1-2.6 and SSP2-4.5), the $PM_{2.5}$ reduction in 2100 is due to emission reductions, while for a high pollution scenario (SSP5-8.5) an approximately equal contribution of emission changes

and climate change to $PM_{2.5}$ mass concentrations for specific world regions (e.g. South America, Asia) was identified. These studies support the validity of our conclusions drawn for pre-industrial times and under the two clean emission scenarios for 2050. However, our results for the regimes in 2050 under SSP3-7.0 may be subject to uncertainties due to neglected climate change effects, although here we focus on the year 2050, when the climate change effects in scenarios of high pollution are smaller than in 2100 (e.g. Fig SPM.8 in IPCC 2021 Summary for Policymakers), and the high emission scenario SSP3-7.0 which we address in this study is cleaner than SSP5-8.5 investigated by Li H. et al. (2022). In summary, the effect of climate change is suggested to be less important than the emission changes for the aerosol regimes investigated in our study. The missing climate change effects might still result in uncertainties in the size and extent of the regimes, but will likely not change their general patterns. Hence, the major conclusions of this study are unlikely to change when climate change is considered.".

*Another point of concern comes from Section 2.3. I can just about understand why present-day clusters cannot be used to analyse regimes at other points in time, but there are two aspects that I do not understand:*

- *First, how does applying a random forest helps? As stated in the introduction, there are few pristine regions in the present day, and they are limited to the aerosols that happen to be emitted in those regions. Don't you necessarily end up extrapolating out of your training dataset when applying the learning to other times?*
- *Second, why did you choose present day as the reference? As stated in the introduction, the reference is normally preindustrial, with some variations as to which year or period is used in practice. And, linking to the previous point, how much does the period used for training matter in terms of regime identification?*

**Response (Major comment #3)**: Thank you for these comments.

Regarding the first point, we performed the clustering in the described way to be able to identify present-day aerosol regimes that show similar conditions as in the pre-industrial case. To achieve comparability between clusters from different time slices we apply a combination of K-means and Random Forest. Random Forest learns the classification criterion from an existing training dataset, i.e. the seven aerosol properties and their regime index under present-day conditions, and then applies the learned criterion to other datasets to find comparable conditions under pre-industrial and future conditions. As stated by the reviewer, this approach could be critical if, for instance, the distribution of pre-industrial regimes was fundamentally different than in the present-day case. To address this issue, we analyzed the regime distribution by K-means for the single time slice PI-1850 (see Fig. below). Although the classified PI-1850 regimes are not directly comparable to our reference training data-set REF-2015, the regime distribution is similar and no new features are emerging, which do not appear in the present-day case.

[Figure]

Regime distribution resulting by the K-means classification for the single time slice PI-1850.

An alternative to the combination of K-means and Random Forest would be to analyze all time slices using a combined dataset with K-means. This approach would also lead to comparable regimes across the time periods, but the drawback would be that the classification results would change whenever a new time period or scenario, that is not included in the present study, is considered.

The reasons for using Random Forest in combination with K-means have been described in the original manuscript in the Introduction (line 99) "The clustering method used in L22, however, was designed for a single time slice and cannot be used for different time periods, as it would lead to incomparable regimes due to the different aerosol conditions in different time periods. More specifically, aerosol conditions during preindustrial times do not agree with the present-day, due to additional contributions from anthropogenic emissions. For our current study, we therefore additionally include the supervised Random Forest machine learning algorithm (Ho 1995, Breiman 2001). As a supervised method, the Random Forest algorithm can be trained using data for one specific time slice and applied to all other time slices. In this way, all time slices are analyzed consistently and the temporal evolution of the aerosol regimes can be studied." and in Sect. 2.3 in the Methodology section (line 208) "The analysis and classification approach applied in this study is based on the K-Means algorithm as in L22 and further extended by the Random Forest algorithm. The latter is required due to the equal variance criterion implemented in the K-means algorithm (Hartigan and Wong, 1979), which would lead to the identification of incomparable regimes across the different time periods when performing K-means classification for each time period independently. Furthermore, applying K-means to a combined dataset of all the different time periods would lead to comparable regimes across all time periods, but the classification results change whenever a new time period or scenario is considered. To overcome these limitations, we developed a two-step approach using a combination of K-means and Random Forest, which is outlined schematically in Fig. 1.".

For better clarity, we have added the following text to the respective paragraph in the Introduction: (line 102) "K-means performs the classification based on individual data variances using an equal variance criterion for classification. Assuming the same number of regimes (K=9) to be generated for present-day and preindustrial times, the variances of the preindustrial regimes and their characteristics would be different from the present-day case due to the different values in the aerosol datasets. This would lead to an incomparability of regimes for different time slices.". A further explanation has been added to Sect. 2.3: (line 210) "… which would divide the datasets based on their individual variances and would therefore lead to …".

Regarding the second point, we note that in contrast to a radiative forcing assessment, where the change in radiative forcing is usually calculated between pre-industrial and present-day, our study compares different aerosol conditions and any time slice can be used as a reference. Since our simulations are nudged towards present-day climate, the simulation of present-day is more accurate than simulations of other time slices, which justifies the choice. Using the preindustrial case as reference might also cause problems, since some present-day conditions do not occur in the pre-industrial case, e.g. the most polluted regimes are missing in pre-industrial times. Moreover, it is unclear which year to use as a preindustrial reference (1750 or 1850).

To clarify this, the following statement has been added to the Methodology section where the present-day (reference) analysis is explained (line 227): "Present-day conditions are selected as our reference for the following reasons. First, our simulations are constrained by present-day climate, so the simulation for the present-day case are likely more accurate than simulations of other time slices. Second, present-day emissions are more reliable than those for past and future conditions. Third, the pre-industrial cases are not fully representative of present-day aerosol conditions, thus using the preindustrial case as a reference, the regime classification for the present-day and 2050 might be incomplete.".

*Other comments:*

*Lines 114-115: Could note that the IPCC AR6 uses 1750 as a preindustrial reference to assess radiative forcing, but uses 1850-1900 for other aspects, like surface temperature change.*

**Response**: Thanks for pointing this out. We have added the following statement to Sect. 2.1 (line 132): "The IPCC Sixth Assessment Report (AR6) uses 1750 as a pre-industrial reference to assess radiative forcing, but uses 1850-1900 for other aspects, e.g. surface temperature change."

*Line 126: "proven" is too strong a word since you do not define what "properly" means. Model skill depends on the level of detail of the comparison, and on the purpose.*

**Response**: Agree, we have replaced "proven" with "shown".

*Line 132: Would be useful to say here that the time slices are 10-year long. That information only appears on line 149, which is a bit late.*

**Response**: Good point. We have added this sentence in Sect. 2.1 (line 152): "The time slices are simulated for a duration of 10-years and the climatological means of the 10-years simulation are considered for the respective time slices."

*Line 149: So learning is done on annual means only? Wouldn't you get more information when using seasonal or monthly means, given the large seasonality of many aerosol types?*

**Response**: Yes, the learning is done on multi-annual means (climatological means).

We indeed investigated the inclusion of seasonal data into the classification for our previous study (L22), but this turned out to be not working, due to the equal-variance criterion applied by the K-means clustering, which also excludes the use of K-means for different time slices and necessitates the introduction of Random Forest in this study as explained above. We refer to L22 for a detailed explanation on why we did not use seasonal data: "Beyond the analysis of multiannual means, we attempted to classify global climatological seasonal data that include the variability in the time dimension concerning the four seasons. This attempt resulted in complications in the classification across the four seasons, since in many cases the seasonal variations are larger than the differences between the specific clusters, which leads to large changes in the characteristics of the clusters and their spatial extent from season to season. This shows that the K-means method discussed here does not work well for analyzing the data variability across time and space simultaneously, as the interpretation of the resulting classification would be challenging. To overcome this limitation, we removed the variability in the time dimension in this study by considering multi-year averages of the model output, thereby setting a focus on classifying the spatial distribution of long-term climatological aerosol regimes. Possible inter-annual and seasonal variability of aerosol properties could alternatively be discussed on the basis of the climatological regimes analyzing the internal temporal changes of aerosol properties within the climatological clusters obtained by K-means".

*Line 208: It would be useful to summarise here the regimes according to L22, and especially what "level" means in, for example, "dust dominated level 1". That information is partly given is section 3.1, which is late.*

**Response**: We thank the reviewer for this suggestion. We have added this information in the Introduction (line 89), as follows: "The lower tropospheric aerosol regimes, as identified in L22, comprise a background regime (occurring in polar regions), two oceanic regimes, with the northern oceanic regime being more polluted than the southern one, two dust regimes, with one being strongly dust dominated and the other representing a mixture of dust and other pollutants, two biomass burning/biogenic regimes, with one comprising fresh aerosol and another one including more aged particles, and three continental regimes including weakly, moderately, and enhanced polluted conditions.".

*Lines 315-317: It would be good to remind the reader that the impact of climate change is not considered here, because this kind of conclusion would probably change if climate feedbacks onto aerosol emissions were included.*

**Response**: Good point. We have added this at the end of the paragraph (line 356): "It needs to be noted that the impact of climate change is not considered here. The possible influence of climate change effects on the presented results is discussed in Sect. 4.".

*Line 361: What is the meaning of having two different regimes for the Arctic and Antarctic? It seems to be purely related to magnitude, rather than changes in composition.*

**Response**: The difference between Arctic and Antarctic can be explained by the lower left panel (boxplot) of Fig. 4. The values of simulated mineral dust, black carbon, sulfate-nitrate-ammonium, particulate organic matter, sea salt and particle number in the accumulation mode are higher in the Arctic than in the Antarctic, but the values of particle number in the Aitken mode is higher in the Antarctic than in the Arctic. This has been added to the text for more clarity (line 406): "A possible explanation for this difference could be the influence of long-range transport of pollutants to the Arctic and new particle formation being favored under the very clean conditions over the Antarctic".

*Line 490-494: That section needs to be more critical of the emission datasets. This is especially true of biomass-burning and biogenic emissions. We do not know what they were in preindustrial conditions (see for example Marlon et al. 2016) and the future climate-driven changes are unlikely to be properly represented in the CMIP emission datasets. Those uncertainties are crucial to some of the conclusions of the paper, for example paragraph 284-299.*

**Response**: We thank the reviewer for this important comment and fully agree with this argument. We have included the following statement to this paragraph (line 568) "Caution is required when using biomass-burning and biogenic emission datasets. A reliable representation of biomass-burning and biogenic emissions during pre-industrial times is not available (e.g. Marlon et al. 2016), and future climate-driven changes are unlikely to be properly represented in the CMIP6 emission inventory used to drive the simulations analysed here. This uncertainty might affect our conclusions regarding biomass-burning and biogenic regimes in terms of their size and extent during pre-industrial times and in the future.".

*Technical comments:*

- *Line 161: evaluating -> evaluation*

**Response**: fixed, thanks for spotting.

**References:**

[revised manuscript text omitted]